# HELIX: Hybrid Encoding with Learnable Identity and Cross-dimensional Synthesis for Time Series Imputation

Fengming Zhang [1 2 3]   Wenjie Du [4]   Huan Zhang [1 2 3]   Ke Yu [1 2 3]   Shen Qu [1 2 3]

## Abstract

Time series imputation benefits from leveraging cross-feature correlations, yet existing attention-based methods re-discover feature relationships at each layer, lacking persistent anchors to maintain consistent representations. To address this, we propose HELIX, which assigns each feature a learnable feature identity, a persistent embedding that captures intrinsic semantic properties throughout the network. Unlike graph-based methods that rely on predefined topology and assume homogeneous spatial relationships, HELIX learns arbitrary feature dependencies end-to-end from temporal co-variation, naturally handling datasets where features mix spatial locations with semantic variables. Integrated with hybrid temporal-feature attention, HELIX achieves the state-of-the-art performance, surpassing all 16 baselines on 5 public datasets across 21 experimental settings in our evaluation. Furthermore, our mechanistic analysis reveals that HELIX aligns learned feature identities and dependencies with latent physical and semantic structure progressively across layers, demonstrating that it more effectively translates cross-feature structure into imputation accuracy.

## 1. Introduction

Missing values in multivariate time-series data, arising from sensor failures, communication dropouts, and irregular sampling, propagate through downstream tasks and degrade performance in forecasting (Wu et al., 2021), classification (Che et al., 2018), and anomaly detection (Wu et al., 2023), especially when missingness spans both time and features.

Deep learning has advanced time series imputation through recurrent methods (Cao et al., 2018; Che et al., 2018), Transformer-based approaches (Du et al., 2023; Nie et al., 2024), and diffusion models (Tashiro et al., 2021; Goswami et al., 2024).

Despite this progress, integrating learnable *feature identity* embeddings with point-wise $(t, i)$ representations for imputation has received limited attention in the imputation literature (Wang et al., 2025; Du et al., 2024; Lester et al., 2021; Devlin et al., 2019). Many graph-based imputation methods couple temporal modeling and cross-feature message passing through a *coarse interface* (Cini et al., 2022; Marisca et al., 2022). A common design strategy is to process one axis first (e.g., summarizing/encoding temporal context before graph propagation, or propagating on feature graphs and then aggregating temporally), which introduces an information bottleneck for *point-wise* reconstruction: collapsing or serializing one dimension can weaken the fine-grained $(t, i)$ alignment required for accurate value imputation under conditions of severe missingness (Cini et al., 2022; Marisca et al., 2022). In addition, when spatial priors are unavailable or heterogeneous feature types coexist, predefined graphs become ambiguous (Cini et al., 2022; Marisca et al., 2022). Moreover, learned adjacency often remains expensive ($O(F^2)$) in practice (Wu et al., 2020; 2019) and still provides no persistent, data-independent anchor when values are missing (Cini et al., 2022). More fundamentally, graph-based formulations assume a homogeneous node set, yet real-world feature dimensions can mix spatial locations with semantic variables or span stations with different variable sets; learning inter-feature relationships purely from temporal co-variation sidesteps these constraints and allows all non-temporal dimensions to be flattened into a unified feature axis without requiring cross-station alignment.

The key insight is that sensor relationships constitute stable structural properties. Therefore, Feature Identity Embedding (FeatID) is introduced: learnable vectors capturing intrinsic semantics, enabling the model to decompose imputation into compatibility (via identity) and dynamic correlation (via values).

[1] Center for Energy and Environmental Policy Research, Beijing Institute of Technology, Beijing 100081, China [2] School of Management, Beijing Institute of Technology, Beijing 100081, China [3] Beijing Lab for System Engineering of Carbon Neutrality, Beijing 100081, China [4] PyPOTS Research. Correspondence to: Shen Qu <squ@bit.edu.cn>, Huan Zhang <zhanghuan19@bit.edu.cn>.

*Proceedings of the 43rd International Conference on Machine Learning*, Seoul, South Korea. PMLR 306, 2026. Copyright 2026 by the author(s).

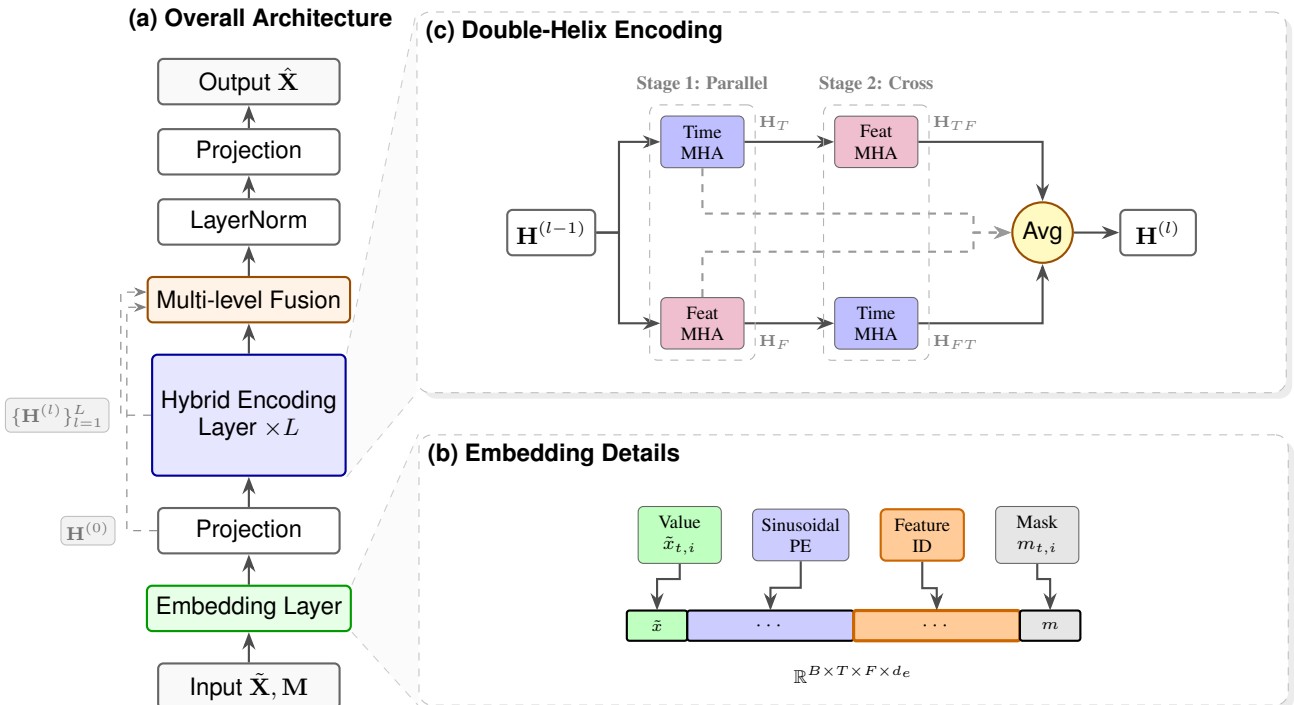

*Figure 1.* Architecture Overview with Zoom-in Details. (a) The main backbone. (b) Embedding details (value, Sinusoidal PE, feature identity, mask). (c) Hybrid Encoding Layer detail (referencing the parallel-then-cross attention mechanism).

HELIX (**H**ybrid **E**ncoding with **L**earnable **I**dentity and Cross-dimensional Synthesis)[1] explicitly decouples static feature identity from dynamic temporal variations. This architecture enriches observations with learned identities and processes them through hybrid encoding that interleaves temporal and cross-feature attention in a double-helix pattern, enabling coordinated information flow across both dimensions.

Our main contributions are:

- **Feature Identity Embedding**: learnable per-feature vectors serving as persistent semantic anchors for cross-feature attention. All inter-feature relationships are discovered implicitly from temporal co-variation, requiring no graph structure or spatial prior. This design allows all non-temporal dimensions to be flattened into a single feature axis without cross-station alignment, accommodating heterogeneous variable sets naturally.
- **Double-Helix architecture**: hybrid encoding yields consistently strong performance across diverse missing patterns and datasets, improving robustness in our ablations (see Tables 4 and 21).
- **State-of-the-art results**: rank first across 21 settings on 5 datasets, outperforming 16 competitive baselines

in our evaluation.
- **Interpretable behavior**: feature attention progressively aligns with underlying structure ($r$: $0.59 \rightarrow 0.71$), embeddings recover both spatial (BeijingAir) and clinical (PhysioNet2012) structure without supervision.

## 2. Related Work

**Deep Multivariate Time Series Imputation.** A recent comprehensive survey categorizes deep learning approaches for imputation into three primary paradigms: predictive, generative, and large model-based methods (Wang et al., 2025).

Predictive methods aim to provide deterministic estimates by modeling temporal dependencies. Early works relied on RNNs with decay mechanisms, such as GRU-D (Che et al., 2018) and BRITS (Cao et al., 2018). Recently, CNN-based models like TimesNet (Wu et al., 2023) and Attention-based models like SAITS (Du et al., 2023) and ImputeFormer (Nie et al., 2024) have gained prominence due to their ability to capture long-range dependencies and complex variations. HELIX belongs to this category, focusing on precise point-wise reconstruction.

Generative methods model the data distribution to quantify uncertainty. Variational Autoencoders (VAEs) like GP-VAE (Fortuin et al., 2020) and GAN-based models like GRUI-GAN (Luo et al., 2018) were early adopters. More re-

---

[1]Code: https://github.com/milaogou/HELIX. HELIX is integrated into PyPOTS https://pypots.com (Du, 2023) for out-of-the-box usage.

cently, diffusion probabilistic models such as CSDI (Tashiro et al., 2021) and PriSTI (Liu et al., 2023) have achieved high-fidelity generation. However, as noted in recent surveys, these methods often suffer from high computational costs and slow inference speeds (Wang et al., 2025).

Large Model-based methods leverage pre-trained foundation models (PFMs) to handle diverse missing patterns. Approaches like GPT4TS (Zhou et al., 2023) and Timer (Liu et al., 2024b) adapt Large Language Models (LLMs) or large-scale pre-trained Transformers for time series tasks. While promising, they pose significant challenges in terms of parameter efficiency and deployment latency compared to specialized architectures.

**Feature Dependency and Graph Modeling.** Capturing correlations between variates is critical for multivariate imputation. GNN-based methods like GRIN (Cini et al., 2022) and SPIN (Marisca et al., 2022) explicitly model these dependencies using graph neural networks. However, they typically rely on predefined spatial topologies or learned adjacency matrices, which assumes a homogeneous node structure. In contrast, HELIX employs a Feature Identity Embedding to learn intrinsic semantic relationships between heterogeneous features without requiring an explicit graph prior, addressing scenarios where spatial information is unavailable or undefined.

**Hybrid Temporal-Feature Encoding.** Multi-scale aggregation has proven effective: TimeMixer (Wang et al., 2024) proposes multi-scale mixing, while iTransformer (Liu et al., 2024a) treats variates as tokens. Crossformer (Zhang & Yan, 2023) also utilizes a two-stage attention mechanism. However, it relies on patch embeddings derived solely from data values. In contrast, HELIX introduces explicit Feature Identity Embeddings, serving as persistent semantic anchors that guide the attention mechanism even when data values are entirely missing.

**Embedding Design in Time Series Transformers.** Effective Transformer performance fundamentally depends on embedding quality: each token must carry sufficient identity information for attention to form meaningful associations. In multivariate time series, a point $(t, i)$ requires anchoring along *both* the temporal and feature axes. Existing approaches typically embed along only one axis. SPIN (Marisca et al., 2022) derives spatial embeddings from an explicit graph, tying feature representations to a predefined topology. ImputeFormer (Nie et al., 2024) learns static per-feature embeddings that serve as soft spatial indices but do not interact with the observation's missingness state. In each case, the embedding is bound to one dimension, so cross-dimensional attention lacks a persistent anchor on the complementary axis. HELIX addresses this by concatenating an independent learnable identity $\mathbf{f}_i$ with sinusoidal encoding $\mathrm{PE}(t)$, value $\tilde{x}_{t,i}$, and mask $m_{t,i}$ *before* projec-

tion, giving every token a dual-anchored representation that remains informative even when observed values are entirely absent.

## 3. Method

### 3.1. Problem Formulation

Let $\mathbf{X} \in \mathbb{R}^{T \times F}$ denote a multivariate time series with $T$ time steps and $F$ features, where $x_{t,i}$ denotes the value at time step $t \in \{1, \ldots, T\}$ and feature $i \in \{1, \ldots, F\}$. Due to missing values, we observe an incomplete version $\tilde{\mathbf{X}}$ along with a binary mask $\mathbf{M} \in \{0, 1\}^{T \times F}$, where $m_{t,i} = 1$ if $x_{t,i}$ is observed and $m_{t,i} = 0$ otherwise. The goal of time series imputation is to learn a function $g_\theta : (\tilde{\mathbf{X}}, \mathbf{M}) \mapsto \hat{\mathbf{X}}$ that accurately reconstructs the complete time series. We construct the observed input by zero-filling missing entries:

$$\tilde{\mathbf{X}} = \mathbf{X} \odot \mathbf{M}, \tag{1}$$

where $\odot$ denotes element-wise multiplication. Thus $\tilde{x}_{t,i} = x_{t,i}$ if $m_{t,i} = 1$, and $\tilde{x}_{t,i} = 0$ otherwise.

### 3.2. Overall Architecture

As shown in Figure 1, HELIX concatenates value, Sinusoidal PE, learnable feature identity, and mask for each $(t, i)$, and projects the result to hidden dimension $d$. It then applies $L$ stacked double-helix hybrid encoding layers, followed by multi-level fusion, LayerNorm, and an output projection to obtain $\hat{\mathbf{X}}$.

### 3.3. Feature Identity Embedding

HELIX introduces Feature Identity Embedding to provide persistent, feature-specific semantics. Since time-series variates lack inherent token identities, we assign each feature a learnable vector so the model can condition cross-feature reasoning on feature type rather than observed values alone.

**Formulation.** Given $F$ features, we introduce a learnable embedding matrix $\mathbf{F}_{\mathrm{id}} \in \mathbb{R}^{F \times d_f}$, where the $i$-th row $\mathbf{f}_i \in \mathbb{R}^{d_f}$ serves as the identity embedding for feature $i \in \{1, \ldots, F\}$. For each observation $(t, i)$, we form an embedding by concatenating four components:

$$\mathbf{e}_{t,i} = \big[ \underbrace{\tilde{x}_{t,i}}_{\text{value}} \, ; \, \underbrace{\mathrm{PE}(t)}_{\text{temporal}} \, ; \, \underbrace{\mathbf{f}_i}_{\text{identity}} \, ; \, \underbrace{m_{t,i}}_{\text{mask}} \big] \in \mathbb{R}^{d_e}. \tag{2}$$

Here $\tilde{x}_{t,i}$ and $m_{t,i}$ are scalars, and $\mathrm{PE}(t) \in \mathbb{R}^{d_{\mathrm{pe}}}$ is a sinusoidal positional encoding. We use the standard sinusoidal encoding (Vaswani et al., 2017):

$$\mathrm{PE}(t)_{2k} = \sin\left(\frac{t}{10000^{2k/d_{\mathrm{pe}}}}\right), \tag{3}$$

$$\mathrm{PE}(t)_{2k+1} = \cos\left(\frac{t}{10000^{2k/d_{\mathrm{pe}}}}\right). \tag{4}$$

Therefore, $d_e = 1 + d_{\mathrm{pe}} + d_f + 1$. This design is inspired by soft prompting in NLP (Lester et al., 2021), where learnable vectors provide task-specific adaptation; here $\mathbf{f}_i$ acts as a feature-specific prompt that conditions the model to process heterogeneous variables distinctly.

**Mechanism: Soft Adjacency Bias.** Let $\mathbf{A} = \mathbf{W}_Q^\top \mathbf{W}_K$. Since $\mathbf{e}_{t,i}$ contains $\mathbf{f}_i$ as a contiguous subvector, we can partition each embedding as $\mathbf{e}_{t,i} = [\mathbf{r}_{t,i}; \mathbf{f}_i]$, where $\mathbf{r}_{t,i}$ collects value, temporal encoding, and mask components. The attention score between features $i$ and $j$ at time $t$ decomposes into four terms:

$$
\begin{aligned}
s_{ij}^{(t)} &= \mathbf{e}_{t,i}^\top \mathbf{A}\, \mathbf{e}_{t,j} \\
&= \underbrace{\mathbf{f}_i^\top \mathbf{A}_{ff}\, \mathbf{f}_j}_{\text{(I) Identity prior}} + \underbrace{\mathbf{f}_i^\top \mathbf{A}_{fr}\, \mathbf{r}_{t,j} + \mathbf{r}_{t,i}^\top \mathbf{A}_{rf}\, \mathbf{f}_j}_{\text{(II) Identity–context cross-terms}} \\
&\quad + \underbrace{\mathbf{r}_{t,i}^\top \mathbf{A}_{rr}\, \mathbf{r}_{t,j}}_{\text{(III) Dynamic context}}\,,
\end{aligned}
\tag{5}
$$

where $\mathbf{A}_{ff}$, $\mathbf{A}_{fr}$, $\mathbf{A}_{rf}$, $\mathbf{A}_{rr}$ are the corresponding subblocks of $\mathbf{A}$.

This decomposition reveals why FeatID is critical under severe missingness. When both $x_{t,i}$ and $x_{t,j}$ are missing (zero-filled), $\mathbf{r}_{t,i}$ reduces to $[0; \mathrm{PE}(t); 0]$, making Term (III) near-degenerate for all feature pairs sharing the same time step. Terms (I) and (II) remain fully informative: Term (I) provides a data-independent compatibility prior via $\mathbf{f}_i^\top \mathbf{A}_{ff} \mathbf{f}_j$, while Terms (II) adaptively modulate this prior using whichever contextual signals are available. Without FeatID, only Term (III) survives, and cross-feature attention collapses under heavy missingness.

**Efficiency and Scalability.** While standard feature attention scales quadratically ($O(TF^2)$), HELIX mitigates this through embedding compression. The required identity dimension scales sub-linearly with feature count (detailed in Section 4.4). For instance, PeMS ($F = 862$) requires only $d_f = 32$, keeping the projection overhead negligible.

**Memory**: Feature Identity Embedding adds only $O(F \cdot d_f)$ parameters, negligible compared to multi-head attention weights $O(d^2)$.

### 3.4. Hybrid Encoding: The Double-Helix Design

After embedding, we obtain $\mathbf{E} \in \mathbb{R}^{B \times T \times F \times d_e}$ and project it to $\mathbf{H}^{(0)} \in \mathbb{R}^{B \times T \times F \times d}$ via a linear layer. We then apply $L$ hybrid encoding layers, each operating on the 4D tensor in two stages that alternate temporal and feature attention streams. The two streams interleave and cross-connect, reminiscent of a DNA double helix, which motivates the name **HELIX**. Stage 1 lets the temporal and feature dimensions refine representations independently, while Stage 2 enables

information exchange via cross-connections. We use *cross-feature* to denote attention over the $F$ features at a fixed time step, and *cross-dimensional* to denote interactions between the temporal and feature dimensions. Notation: within the $l$-th encoding layer, we denote the four branch outputs by $\mathbf{H}_T^{(l)}$, $\mathbf{H}_F^{(l)}$, $\mathbf{H}_{TF}^{(l)}$, and $\mathbf{H}_{FT}^{(l)}$, and the fused layer output by $\mathbf{H}^{(l)}$.

**Stage 1: Dimension-specific Decoupling.** Multi-head temporal and cross-feature attention are applied independently and in parallel:

$$
\mathbf{H}_T^{(l)} = \text{Temporal MHA}(\mathbf{H}^{(l-1)}) \tag{6}
$$

$$
\mathbf{H}_F^{(l)} = \text{Feature MHA}(\mathbf{H}^{(l-1)}) \tag{7}
$$

Temporal attention reshapes to $\mathbb{R}^{(B \cdot F) \times T \times d}$ and applies attention over $T$; feature attention reshapes to $\mathbb{R}^{(B \cdot T) \times F \times d}$ and applies attention over $F$. Both include LayerNorm and residual connections.

**Stage 2: Inter-dimensional Synthesis.** To enable information exchange between temporal and feature dimensions, serial cross-attention is employed:

$$
\mathbf{H}_{TF}^{(l)} = \text{Feature MHA}(\mathbf{H}_T^{(l)}) \tag{8}
$$

$$
\mathbf{H}_{FT}^{(l)} = \text{Temporal MHA}(\mathbf{H}_F^{(l)}) \tag{9}
$$

**Layer-wise Fusion.** Within each layer, these four outputs are fused:

$$
\mathbf{H}^{(l)} = \frac{1}{4}\left(\mathbf{H}_T^{(l)} + \mathbf{H}_F^{(l)} + \mathbf{H}_{TF}^{(l)} + \mathbf{H}_{FT}^{(l)}\right) \tag{10}
$$

### 3.5. Multi-level Fusion

We aggregate representations from all encoding stages, omitting $\mathbf{H}^{(l)}$ since it is a linear combination of them. This avoids double-counting because $\mathbf{H}^{(l)}$ is the average of the four branch outputs in Eq. (10):

$$
\tilde{\mathbf{H}} = \frac{1}{1+4L}\left(\mathbf{H}^{(0)} + \sum_{l=1}^{L}\left(\mathbf{H}_T^{(l)} + \mathbf{H}_F^{(l)} + \mathbf{H}_{TF}^{(l)} + \mathbf{H}_{FT}^{(l)}\right)\right) \tag{11}
$$

Empirical results indicate that learnable gated fusion underperforms simple averaging (Appendix Section D). This observation aligns with findings in deep residual learning (He et al., 2016), where direct identity mappings facilitate better signal propagation and gradient flow compared to complex gating mechanisms, ensuring that information from all abstraction levels remains accessible. The aggregated representation is then normalized and projected:

$$
\hat{\mathbf{X}} = \text{Linear}(\text{LayerNorm}(\tilde{\mathbf{H}})) \tag{12}
$$

*Table 1.* Overall ranking across all experimental settings. Lower average rank indicates better performance. † indicates models that could not run on all settings due to computational or architectural constraints.

| Model | Avg. Rank ↓ | Valid Exps. | Global Rank | Category | Venue |
|---|---|---|---|---|---|
| **HELIX (Ours)** | **1.00** | 21/21 | **1** | Ours | ICML'26 |
| ImputeFormer | 3.29 | 21/21 | 2 | Low-rank Attention | KDD'24 |
| SAITS | 3.76 | 21/21 | 3 | Masked Attention | ESWA'23 |
| StemGNN | 5.71 | 21/21 | 4 | Graph Neural Network | NeurIPS'20 |
| Linear Interpolation | 6.67 | 21/21 | 5 | Naive | – |
| PatchTST | 7.24 | 21/21 | 6 | Patch-based | ICLR'23 |
| Nonstationary Trans. | 7.33 | 21/21 | 7 | Non-stationary Attn | NeurIPS'22 |
| FreTS | 7.48 | 21/21 | 8 | Frequency Domain | NeurIPS'23 |
| iTransformer | 7.95 | 21/21 | 9 | Variate Attention | ICLR'24 |
| TEFN | 8.67 | 21/21 | 10 | Evidence Fusion | TPAMI'25 |
| Time-LLM† | 11.75 | 16/21 | 11 | LLM Adaptation | ICLR'24 |
| TimeMixer | 11.86 | 21/21 | 12 | Multi-scale Mixing | ICLR'24 |
| LOCF | 12.05 | 21/21 | 13 | Naive | – |
| ModernTCN | 12.43 | 21/21 | 14 | Modern Convolution | ICLR'24 |
| TimeMixer++† | 13.06 | 16/21 | 15 | Multi-scale Mixing | ICLR'25 |
| TOTEM | 14.38 | 21/21 | 16 | Tokenization | TMLR'24 |
| MOMENT† | 16.82 | 11/21 | 17 | Foundation Model | ICML'24 |

### 3.6. Training Objective

The model is trained under each missingness pattern, and we report the results under the same pattern-specific protocol at test time. Let $\mathcal{O} = \{(t,i) \mid m_{t,i} = 1\}$ denote observed indices. Following SAITS (Du et al., 2023), we further randomly mask a subset of observed entries during training, denoted by $\mathcal{M}_{\text{art}} \subset \mathcal{O}$ (artificially masked), and compute imputation MIT on the artificially masked entries; optionally also compute ORT on the remaining observed entries. Note that $\tilde{\mathbf{X}}$ is the model input, while $x_{t,i}$ denotes the ground-truth value used for supervision/evaluation.

**Observed Reconstruction Task (ORT):**

$$\mathcal{L}_{\text{ORT}} = \frac{1}{|\mathcal{O} \setminus \mathcal{M}_{\text{art}}|} \sum_{(t,i) \in \mathcal{O} \setminus \mathcal{M}_{\text{art}}} |x_{t,i} - \hat{x}_{t,i}| \qquad (13)$$

**Masked Imputation Task (MIT):**

$$\mathcal{L}_{\text{MIT}} = \frac{1}{|\mathcal{M}_{\text{art}}|} \sum_{(t,i) \in \mathcal{M}_{\text{art}}} |x_{t,i} - \hat{x}_{t,i}| \qquad (14)$$

The total loss combines Observed Reconstruction Task (ORT) and Masked Imputation Task (MIT): $\mathcal{L} = \mathcal{L}_{\text{ORT}} + \mathcal{L}_{\text{MIT}}$. Equal weights are assigned to both terms.

## 4. Experiments

### 4.1. Experimental Setup

**Datasets.** We evaluate on five real-world datasets covering diverse domains: **PhysioNet2012** (Silva et al., 2012) (ICU vital signs, 48 steps, 35 features), **BeijingAir** (Zhang et al., 2017) (air quality, 24 steps, 132 features), **ItalyAir** (Vito,

2016) (air quality, 12 steps, 13 features), **ETT-h1** (Zhou et al., 2021) (electricity transformer, 48 steps, 7 features), and **PeMS** (Chen et al., 2001) (traffic sensors, 24 steps, 862 features).

**Missing Patterns.** Experiments adopt the TSI-Bench (Du et al., 2024) protocols: Point-X% (X ∈ 10, 50, 90), Block-50%, and Subseq-50%.

**Data Splits.** Standard PyPOTS (Du, 2023) splits are used. Following SAITS (Du et al., 2023) and PyPOTS defaults, ORT and MIT losses are equally weighted across all models to ensure fair comparison.

**Baselines.** We compare HELIX against competitive methods selected from TSI-Bench (Du et al., 2024) along with several recent methods, spanning attention-based, GNN, convolution, and foundation model architectures (see Table 1). Diffusion-based generative models are omitted because TSI-Bench results (Du et al., 2024) confirm that leading diffusion baselines exhibit substantially higher cross-dataset variance and inference cost than predictive methods under these protocols. For reference, on BeijingAir Point-10%, HELIX achieves MAE $0.073 \pm 0.004$ compared to CSDI's $0.102 \pm 0.010$ reported in TSI-Bench, a 28.4% improvement with $171\times$ faster inference.

**Implementation.** All experiments are conducted using the PyPOTS (Du, 2023) framework to ensure reproducibility and fair comparison. We perform 25 HPO trials per model–dataset pair and repeat training with 5 random seeds for reporting mean±std. For significance testing, we additionally repeat the same best configuration with 25 seeds on ETT-h1 Point-50%. HELIX hyperparameters are tuned per dataset (see Table 26 in Appendix); typical ranges: $d_{pe} \in [6, 24]$, $d_f \in [6, 32]$, $d \in [32, 576]$, $L \in [2, 3]$.

*Table 2.* Detailed MAE results on ETT-h1 (48 steps, 7 features) across all missing patterns. Mean ± std over 5 runs. Ranking: **1st**, 2nd; Time reports wall-clock inference time for imputing the test set (per run).

| Method | Point-10% | Point-50% | Point-90% | Block-50% | Subseq-50% | #Params↓ | Time↓ |
|---|---|---|---|---|---|---|---|
| Mean Imputation | 0.737±N/A | 0.738±N/A | 0.739±N/A | 0.720±N/A | 0.773±N/A | N/A | N/A |
| Median Imputation | 0.710±N/A | 0.708±N/A | 0.715±N/A | 0.712±N/A | 0.784±N/A | N/A | N/A |
| LOCF | 0.315±N/A | 0.425±N/A | 0.763±N/A | 0.721±N/A | 0.809±N/A | N/A | N/A |
| Linear Interpolation | 0.197±N/A | 0.267±N/A | 0.616±N/A | 0.527±N/A | 0.722±N/A | N/A | N/A |
| HELIX (Ours) | **0.128±0.005** | **0.189±0.012** | **0.429±0.011** | **0.372±0.015** | **0.489±0.014** | 803.5K | 0.06s |
| ImputeFormer | 0.202±0.044 | 0.296±0.036 | 0.492±0.005 | 0.404±0.021 | 0.520±0.017 | 124.0K | 0.08s |
| SAITS | 0.150±0.007 | 0.208±0.009 | 0.440±0.016 | 0.422±0.019 | 0.620±0.016 | 88.2M | 0.05s |
| Nonstationary Trans. | 0.301±0.009 | 0.358±0.007 | 0.510±0.007 | 0.483±0.007 | 0.586±0.009 | 589.9K | 0.01s |
| PatchTST | 0.229±0.016 | 0.272±0.027 | 0.503±0.009 | 0.529±0.008 | 0.689±0.030 | 72.2K | 0.01s |
| iTransformer | 0.269±0.003 | 0.339±0.003 | 0.594±0.006 | 0.494±0.004 | 0.740±0.014 | 23.7M | 0.02s |
| TEFN | 0.307±0.003 | 0.475±0.019 | 0.622±0.015 | 0.576±0.015 | 0.672±0.015 | 985 | 0.01s |
| TimeMixer | 0.578±0.231 | 0.699±0.121 | 0.835±0.009 | 0.768±0.039 | 0.877±0.011 | 11.1K | 0.02s |
| TimeMixer++ | 0.333±0.007 | 0.378±0.006 | 0.593±0.002 | 0.515±0.008 | 0.665±0.008 | 11.2M | 0.20s |
| ModernTCN | 0.298±0.011 | 0.435±0.012 | 0.786±0.014 | 0.593±0.010 | 0.771±0.005 | 175.7K | 0.03s |
| StemGNN | 0.277±0.005 | 0.314±0.014 | 0.521±0.023 | 0.426±0.011 | 0.611±0.009 | 1.6M | 0.02s |
| TOTEM | 0.368±0.195 | 0.526±0.026 | 0.817±0.018 | 0.741±0.009 | 0.868±0.002 | 23.8K | 0.03s |
| FreTS | 0.264±0.020 | 0.294±0.022 | 0.540±0.014 | 0.505±0.014 | 0.661±0.019 | 465.3K | 0.01s |
| Time-LLM | 0.290±0.001 | 0.417±0.030 | 0.567±0.010 | 0.614±0.004 | 0.775±0.004 | 31.1M | 3.81s |
| MOMENT | 0.550±0.145 | 0.757±0.129 | 0.853±0.014 | 0.838±0.013 | 0.924±0.026 | 109.7M | 3.82s |

## 4.2. Main Results

We compute the rank within each experimental setting by sorting methods by MAE (lower is better), and report the average rank across all settings. For methods marked with † that could not be evaluated on some settings (e.g., sequence-length or memory constraints), we report the average rank over successfully completed settings and also list the number of valid experiments. As shown in Table 1, HELIX ranks 1st in all 21 settings.

**Key findings.** Linear Interpolation outperforms several deep methods, yet HELIX achieves +37.7% improvement over it. On PhysioNet2012, HELIX achieves 41% improvement, demonstrating clinical relevance. HELIX also maintains parameter efficiency: 803K vs. SAITS (88M) and iTransformer (23.7M) on ETT-h1 setting. HELIX's advantage is consistent across metrics, achieving the best average rank on MAE, MSE, and MRE (Appendix Section C).

Table 2 provides detailed MAE comparison on ETT-h1 across all missing patterns. HELIX consistently achieves top-3 performance across most settings, with particularly strong results on challenging Block-50% and Subseq-50% patterns. Complete results for all five datasets are provided in Appendix Section B.

**Statistical Significance.** Wilcoxon signed-rank tests (25 runs, ETT-h1 Point-50%) confirm HELIX significantly outperforms all baselines ($p < 0.001$, Table 3). While individual ablation variants occasionally match or surpass HELIX on specific settings (e.g., w/o Fusion on ETT-h1, w/o Hybrid on BeijingAir Point-90%), none achieves consistently top performance across all patterns and datasets (Tables 4

*Table 3.* Wilcoxon signed-rank tests (ETT-h1, Point-50%, 25 independent seeds). HELIX significantly outperforms all baselines ($p < 0.001$).

| Comparison | HELIX | Baseline | Diff | *p*-value |
|---|---|---|---|---|
| vs ImputeFormer | .189±.012 | .296±.036 | -.107 | <0.001 |
| vs SAITS | .189±.012 | .208±.009 | -.019 | <0.001 |
| vs PatchTST | .189±.012 | .272±.027 | -.083 | <0.001 |
| vs iTransformer | .189±.012 | .339±.003 | -.150 | <0.001 |
| vs w/o FeatID | .189±.012 | .241±.012 | -.052 | <0.001 |

and 21): removing Hybrid Encoding causes catastrophic failure on Subseq-50% (0.294 vs. 0.166 on BeijingAir), and removing Feature Identity degrades all settings substantially. Only the full HELIX ranks first across all 21 settings, indicating that each component contributes to cross-pattern robustness rather than peak single-setting accuracy. Notably, removing Feature Identity Embedding causes significant degradation ($p < 0.001$), underscoring its essential role.

## 4.3. Ablation Study

We evaluate four ablation variants (detailed in Appendix Section F): removing Multi-level Fusion (use only final layer), replacing sinusoidal with learnable Sinusoidal PE, removing Hybrid Encoding (pure serial Time→Feature→Time), and removing Feature Identity Embedding entirely.

Table 4 presents ablation results on BeijingAir with per-pattern breakdown. Complete ablation results across all datasets and missing patterns are provided in Table 21.

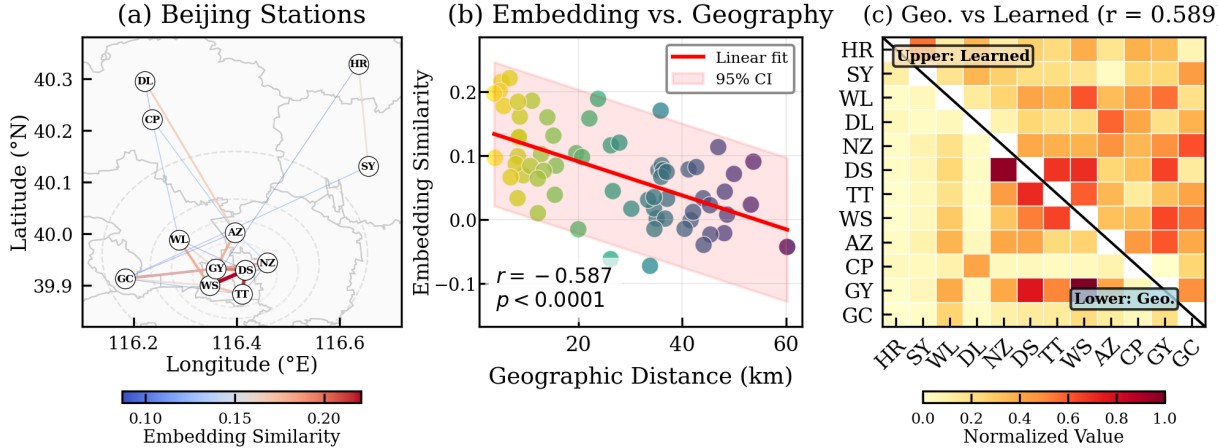

*Figure 2.* Learned Feature Identity Embeddings on BeijingAir. (a) Geographic distribution with the top 25 learned connections. (b) Embedding similarity vs. geographic distance ($r = -0.587$, $p < 0.0001$). (c) Comparison: learned similarity (upper) vs. geographic proximity (lower). Feature Identity Embedding implicitly learns spatial structure without explicit graph modeling. Station abbreviations: HR=Huairou, SY=Shunyi, WL=Wanliu, DL=Dingling, NZ=Nongzhanguan, DS=Dongsi, TT=Tiantan, WS=Wanshouxigong, AZ=Aotizhongxin, CP=Changping, GY=Guanyuan, GC=Gucheng.

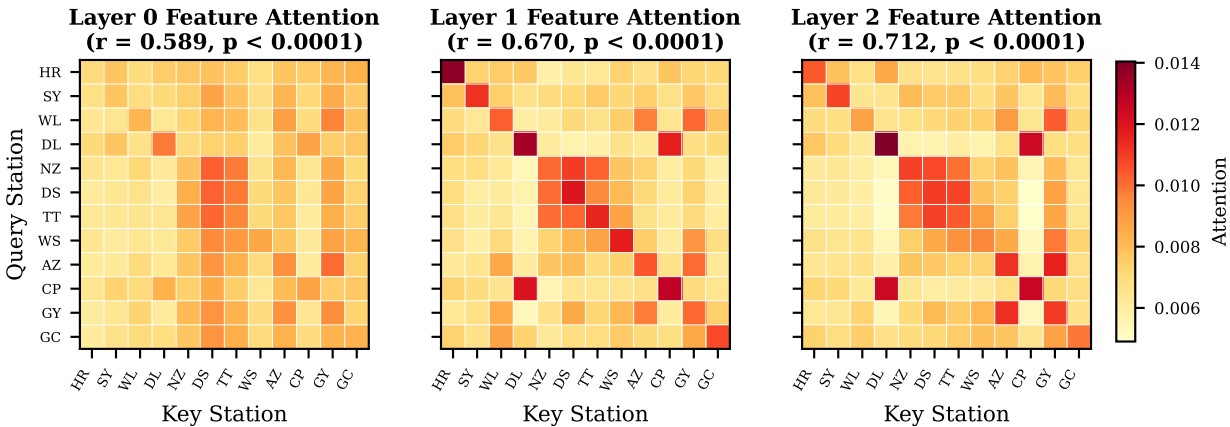

*Figure 3.* Feature attention increasingly captures spatial structure across layers. Correlation with geographic proximity: Layer 0 ($r = 0.589$), Layer 1 ($r = 0.670$), Layer 2 ($r = 0.712$), all $p < 0.0001$.

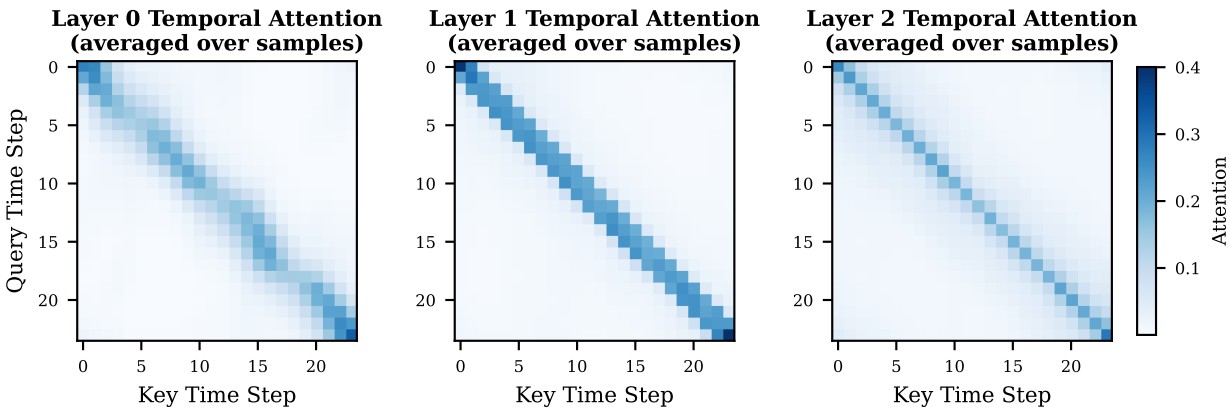

*Figure 4.* Evolution of temporal attention patterns across layers on BeijingAir. **Layer 0**: Diffuse attention along the diagonal with gradual decay. **Layer 1**: Sharp concentration on immediately adjacent time steps. **Layer 2**: Balanced pattern combining local focus with broader context. We interpret this progression as *perceiving→focusing→understanding*, suggesting hierarchical temporal abstraction.

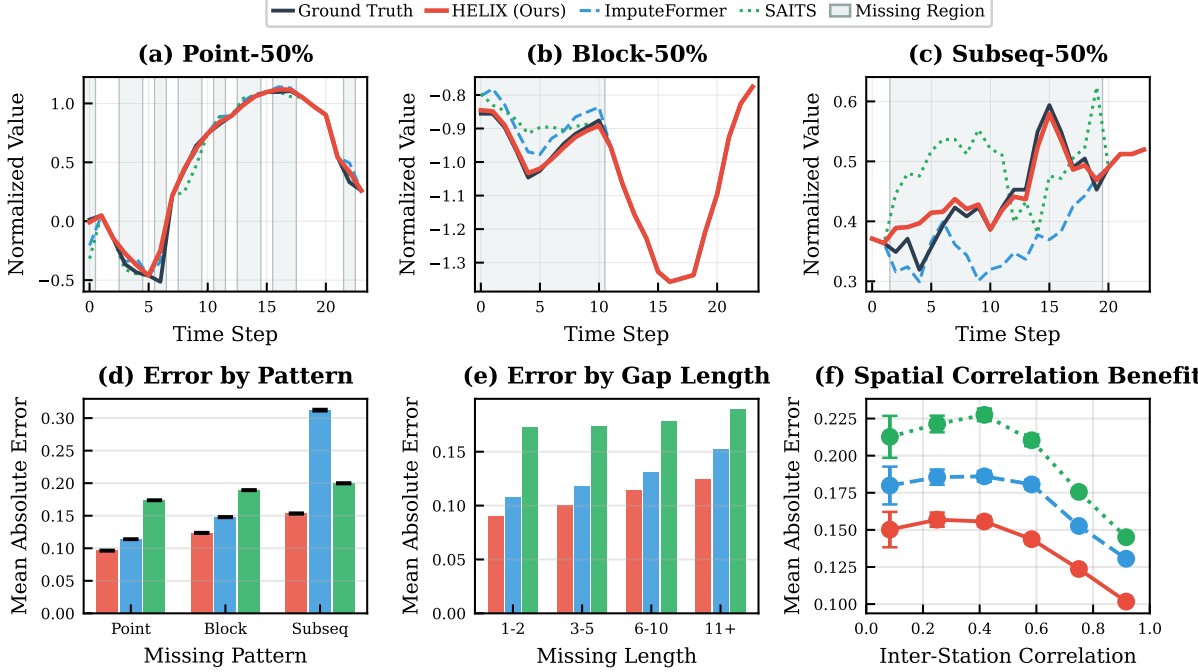

*Figure 5.* Qualitative and quantitative comparison of imputation results on BeijingAir. **(a)–(c)** Time series visualization across three missing patterns, with gray regions indicating missing values. HELIX (red) tracks the ground truth most closely, especially at pattern transitions. **(d)** Mean Absolute Error comparison confirms HELIX's consistent advantage across all patterns. **(e)** Error increases with gap length for all methods, but HELIX maintains lower error even for long gaps (11+ steps), demonstrating effective long-range dependency modeling. **(f)** Key finding: HELIX's error decreases more steeply with cross-station correlation than baselines, demonstrating stronger exploitation of cross-feature structure.

*Table 4.* Ablation on BeijingAir across missing patterns (MAE). Ranking per column among ablations: **1st**, 2nd.

| Model | Point-50% | Block-50% | Subseq-50% |
|---|---|---|---|
| **HELIX (Ours)** | **0.102±0.005** | **0.131±0.005** | **0.166±0.009** |
| w/o Fusion | 0.104±0.006 | 0.147±0.019 | 0.173±0.014 |
| w/o Sinusoidal | 0.108±0.002 | 0.142±0.003 | 0.173±0.010 |
| w/o Hybrid | 0.104±0.004 | 0.137±0.004 | 0.294±0.101 |
| w/o FeatEmb | 0.144±0.006 | 0.223±0.009 | 0.398±0.016 |

*Table 5.* Feature Identity Embedding dimension scaling. Large-scale spatially-structured datasets (PeMS) achieve high compression (27:1). ETT-h1 shows expansion (0.6:1) because its 7 features lack inherent structure, requiring additional embedding capacity to learn implicit relationships.

| Dataset | #Features | $d_f$ | Ratio |
|---|---|---|---|
| PeMS | 862 | 32 | 27:1 |
| BeijingAir | 132 | 24 | 5.5:1 |
| PhysioNet2012 | 35 | 16 | 2.2:1 |
| ItalyAir | 13 | 6 | 2.2:1 |
| ETT-h1 | 7 | 12 | 0.6:1 |

**Component contributions.** Table 4 shows that HELIX achieves the best overall performance on BeijingAir across five missing patterns. Across datasets, we observe that different components contribute differently depending on the domain and missingness type (see Table 21).

**Multi-level fusion improves worst-case robustness.** Removing Multi-level Fusion systematically improves MAE on smaller datasets (ETT-h1 and ItalyAir across most patterns; Tables 22 and 23). However, Table 24 reveals a clear asymmetry on PeMS: when fusion underperforms, the gap is modest (Point-10%: 0.097 vs. 0.095, ~2%); when it helps, the gain is substantial (Subseq-50%: 0.311 vs. 0.542, ~74%). This asymmetry reflects an evaluation framing issue rather than a design flaw: in realistic deployments, mul-

tiple missingness patterns co-occur simultaneously within the same dataset, so robustness across all patterns matters more than single-pattern optimization. Across all 21 settings, full HELIX with multi-level fusion achieves the best overall rank.

**Feature Identity is consistently essential.** Removing Feature Identity Embedding causes the most consistent and substantial degradation across all datasets and missing patterns (Table 21), confirming that learnable identities provide stable semantic anchors for cross-feature reasoning, especially under heavy missingness.

**Hybrid encoding and Sinusoidal positional encoding have complementary effects.** Ablations on Hybrid Encoding and Sinusoidal positional encoding can yield occasional improvements on specific settings, but they more frequently degrade performance on long-gap patterns (e.g., Subseq-50%), suggesting these components mainly benefit long-range dependency modeling and cross-dimensional information exchange (Tables 4 and 21).

### 4.4. Analysis and Visualization

**Emergent Structure Discovery.** On BeijingAir, where geographic coordinates are never provided, embedding similarity strongly anticorrelates with physical distance ($r = -0.587$, $p < 0.0001$, Figure 2), demonstrating that HELIX reconstructs spatial structure purely from temporal co-variation. On PhysioNet2012, clinically related features cluster together ($p < 0.001$; Appendix Section E), confirming this structure discovery generalizes beyond spatial domains.

**Progressive Attention Refinement.** Correlation with geographic proximity increases with depth (Figure 3): $r = 0.589 \rightarrow 0.670 \rightarrow 0.712$ across layers.

**Temporal Attention Evolution.** Figure 4 reveals a three-stage hierarchical learning process. Layer 0 shows diffuse diagonal attention (*perceiving*), Layer 1 exhibits sharp concentration on adjacent timesteps (*focusing*), and Layer 2 achieves a balanced local-global pattern (*understanding*). This temporal hierarchy parallels the spatial refinement in Figure 3, suggesting HELIX learns coordinated multi-scale abstractions across both dimensions.

**Embedding Dimension Efficiency.** Optimal $d_f$ scales sublinearly with feature count (Table 5). For instance, PeMS ($F = 862$) achieves 27:1 compression with $d_f = 32$, minimizing parameter overhead. Conversely, low-dimensional datasets like ETT-h1 ($F=7$) benefit from expansion ($d_f > F$), utilizing additional capacity to capture implicit relationships where inherent structure is sparse.

**Structure-to-Accuracy Translation.** HELIX's performance advantage amplifies with cross-feature correlation (Figure 5(f)), with gains over ImputeFormer rising from 16.5% (low correlation) to 22.1% (high correlation). Notably, even in the lowest correlation bin, HELIX still achieves a 32.3% error reduction over the naive baseline, comparable to SAITS's 31.8%, confirming that FeatID does not sacrifice low-correlation performance to boost high-correlation cases. This confirms that Feature Identity Embedding effectively exploits latent structure while remaining competitive where cross-feature dependencies are weak. Additionally, HELIX exhibits superior robustness to gap length (Figure 5(e)), maintaining lower error rates even for missing sequences exceeding 11 steps.

## 5. Conclusion

Feature Identity Embedding yields decisive improvements for multivariate time series imputation. Its necessity is established by three complementary lines of evidence: ablation degradation ($p < 0.001$), unsupervised recovery of latent structure (geographic and clinical), and progressive attention alignment with underlying dependencies across layers. More broadly, we believe the most transferable insight from this work is about embedding design rather than any specific architectural choice. Whether a Transformer can realize its potential in a given application depends, to a large extent, on the quality of the representation each token carries before it enters the attention mechanism. Equipping every token with sufficient identity information, so that the model can distinguish what a token represents independently of the observed value, matters more than stacking additional layers or introducing more sophisticated attention variants. Our experiments support this claim quantitatively: simply adding Feature Identity Embedding to a standard Transformer yields larger gains than switching to more complex architectures that lack such embedding. This principle is not specific to time series imputation; it applies to any scenario that requires cross-variate reasoning, including forecasting, anomaly detection, and sensor fusion, wherever multiple heterogeneous variables must inform each other through attention.

**Limitations.** HELIX learns embeddings per dataset; cross-dataset transfer and scaling to $F > 10^3$ require further investigation.

## Impact Statement

This paper advances time series analysis with applications in healthcare, environmental science, and urban computing. While improved imputation enables better decision-making, practitioners should treat imputed values as estimates with appropriate uncertainty quantification in critical applications.

## Acknowledgements

Shen Qu is thankful for support from the National Science Fund for Distinguished Young Scholars of China (52425005) and General Program of National Natural Science Foundation of China (52370189).

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

# A. Notation Summary

*Table 6.* Summary of mathematical notation used in this paper.

| Symbol | Description |
|---|---|
| $T$ | Number of time steps |
| $F$ | Number of features |
| $B$ | Batch size |
| $L$ | Number of hybrid encoding layers |
| $H$ | Number of attention heads |
| $\mathbf{X} \in \mathbb{R}^{T \times F}$ | Input time series |
| $\tilde{\mathbf{X}} \in \mathbb{R}^{T \times F}$ | Observed input after zero-filling ($\tilde{\mathbf{X}} = \mathbf{X} \odot \mathbf{M}$) |
| $x_{t,i}$ | Value at time step $t$, feature $i$ |
| $\tilde{x}_{t,i}$ | Input value at $(t, i)$ (equals $x_{t,i}$ if observed, else 0) |
| $\mathbf{M} \in \{0, 1\}^{T \times F}$ | Missingness mask (1=observed, 0=missing) |
| $\hat{\mathbf{X}}$ | Imputed time series |
| $d$ | Model hidden dimension |
| $d_e$ | Embedding dimension ($d_e = 1 + d_{pe} + d_f + 1$) |
| $d_{pe}$ | Temporal positional encoding dimension |
| $d_f$ | Feature identity embedding dimension |
| $d_k$ | Per-head attention dimension ($d_k = d/H$) |
| $\mathbf{F}_{\text{id}} \in \mathbb{R}^{F \times d_f}$ | Feature identity embedding matrix |
| $\mathbf{f}_i \in \mathbb{R}^{d_f}$ | Identity embedding for feature $i$ |
| $\mathbf{e}_{t,i} \in \mathbb{R}^{d_e}$ | Combined embedding for observation $(t, i)$ |
| $\mathbf{H}^{(l)}$ | Hidden representation at layer $l$ |
| $\mathbf{H}_T^{(l)}, \mathbf{H}_F^{(l)}$ | Outputs of temporal/feature attention (Stage 1) |
| $\mathbf{H}_{TF}^{(l)}, \mathbf{H}_{FT}^{(l)}$ | Outputs of cross-dimensional attention (Stage 2) |

# B. Complete Experimental Results

We report complete MAE results for all 21 experimental settings. Best results are **bolded**, second best are underlined.

## B.1. Results on BeijingAir

*Table 7.* Complete MAE results on BeijingAir across all missing patterns. Mean ± std over 5 runs. Ranking: **1st**, 2nd.

| Model | Point-10% | Point-50% | Point-90% | Block-50% | Subseq-50% | #Params↓ | Time↓ |
|---|---|---|---|---|---|---|---|
| *Naive Baselines* | | | | | | | |
| Linear Interpolation | 0.112±N/A | 0.165±N/A | 0.366±N/A | 0.285±N/A | 0.358±N/A | N/A | N/A |
| LOCF | 0.188±N/A | 0.264±N/A | 0.500±N/A | 0.400±N/A | 0.482±N/A | N/A | N/A |
| Median Imputation | 0.681±N/A | 0.677±N/A | 0.680±N/A | 0.682±N/A | 0.676±N/A | N/A | N/A |
| Mean Imputation | 0.721±N/A | 0.708±N/A | 0.716±N/A | 0.714±N/A | 0.711±N/A | N/A | N/A |
| *Deep Learning Methods* | | | | | | | |
| **HELIX (Ours)** | **0.073±0.004** | **0.102±0.005** | **0.190±0.005** | **0.131±0.005** | **0.166±0.009** | 4.6M | 2.44s |
| ImputeFormer | 0.095±0.005 | 0.122±0.002 | 0.245±0.004 | 0.158±0.004 | 0.322±0.009 | 3.4M | 0.56s |
| SAITS | 0.151±0.001 | 0.190±0.003 | 0.312±0.015 | 0.207±0.005 | 0.217±0.005 | 7.2M | 0.06s |
| Nonstationary Trans. | 0.178±0.003 | 0.201±0.002 | 0.355±0.001 | 0.275±0.003 | 0.331±0.009 | 7.0M | 0.05s |
| PatchTST | 0.181±0.002 | 0.214±0.005 | 0.297±0.001 | 0.273±0.002 | 0.301±0.003 | 30.3M | 4.68s |
| iTransformer | 0.124±0.004 | 0.160±0.003 | 0.353±0.003 | 0.349±0.061 | 0.626±0.010 | 8.3M | 0.13s |
| TEFN | 0.141±0.004 | 0.198±0.001 | 0.387±0.001 | 0.315±0.001 | 0.353±0.001 | 37.5K | 0.04s |
| TimeMixer | 0.278±0.007 | 0.293±0.010 | 0.327±0.003 | 0.300±0.007 | 0.311±0.003 | 168.7K | 0.32s |
| TimeMixer++ | 0.532±0.061 | 0.615±0.020 | 0.731±0.013 | 0.734±0.010 | 0.738±0.010 | 33.5M | 9.82s |
| ModernTCN | 0.257±0.012 | 0.371±0.017 | 0.716±0.004 | 0.492±0.020 | 0.590±0.014 | 14.1M | 0.23s |
| StemGNN | 0.170±0.004 | 0.185±0.003 | 0.249±0.002 | 0.233±0.007 | 0.270±0.007 | 1.1M | 0.49s |
| TOTEM | 0.322±0.206 | 0.452±0.013 | 0.743±0.002 | 0.682±0.007 | 0.725±0.015 | 98.9K | 0.08s |
| FreTS | 0.193±0.006 | 0.217±0.005 | 0.265±0.022 | 0.260±0.016 | 0.294±0.002 | 909.9K | 0.04s |
| Time-LLM | 0.211±0.005 | 0.223±0.013 | 0.385±0.006 | 0.469±0.055 | 0.607±0.024 | 31.8M | 221.62s |
| MOMENT | 0.607±0.650 | 1.383±1.247 | 0.740±0.010 | 0.854±0.039 | 0.799±0.221 | 109.8M | 10.91s |

## B.2. Results on ETT-h1

*Table 8.* Complete MAE results on ETT_h1 across all missing patterns. Mean ± std over 5 runs. Ranking: **1st**, 2nd.

| Model | Point-10% | Point-50% | Point-90% | Block-50% | Subseq-50% | #Params↓ | Time↓ |
|---|---|---|---|---|---|---|---|
| *Naive Baselines* | | | | | | | |
| Linear Interpolation | 0.197±N/A | 0.267±N/A | 0.616±N/A | 0.527±N/A | 0.722±N/A | N/A | N/A |
| LOCF | 0.315±N/A | 0.425±N/A | 0.763±N/A | 0.721±N/A | 0.809±N/A | N/A | N/A |
| Median Imputation | 0.710±N/A | 0.708±N/A | 0.715±N/A | 0.712±N/A | 0.784±N/A | N/A | N/A |
| Mean Imputation | 0.737±N/A | 0.738±N/A | 0.739±N/A | 0.720±N/A | 0.773±N/A | N/A | N/A |
| *Deep Learning Methods* | | | | | | | |
| **HELIX (Ours)** | **0.128±0.005** | **0.189±0.012** | **0.429±0.011** | **0.372±0.015** | **0.489±0.014** | 803.5K | 0.06s |
| ImputeFormer | 0.202±0.044 | 0.296±0.036 | 0.492±0.005 | 0.404±0.021 | 0.520±0.017 | 124.0K | 0.08s |
| SAITS | 0.150±0.007 | 0.208±0.009 | 0.440±0.016 | 0.422±0.019 | 0.620±0.016 | 88.2M | 0.05s |
| Nonstationary Trans. | 0.301±0.009 | 0.358±0.007 | 0.510±0.007 | 0.483±0.007 | 0.586±0.009 | 589.9K | 0.01s |
| PatchTST | 0.229±0.016 | 0.272±0.027 | 0.503±0.009 | 0.529±0.008 | 0.689±0.030 | 72.2K | 0.01s |
| iTransformer | 0.269±0.003 | 0.339±0.003 | 0.594±0.006 | 0.494±0.004 | 0.740±0.014 | 23.7M | 0.02s |
| TEFN | 0.307±0.003 | 0.475±0.019 | 0.622±0.015 | 0.576±0.015 | 0.672±0.015 | 985 | 0.01s |
| TimeMixer | 0.578±0.231 | 0.699±0.121 | 0.835±0.009 | 0.768±0.039 | 0.877±0.011 | 11.1K | 0.02s |
| TimeMixer++ | 0.333±0.007 | 0.378±0.006 | 0.593±0.002 | 0.515±0.008 | 0.665±0.008 | 11.2M | 0.20s |
| ModernTCN | 0.298±0.011 | 0.435±0.012 | 0.786±0.014 | 0.593±0.010 | 0.771±0.005 | 175.7K | 0.03s |
| StemGNN | 0.277±0.005 | 0.314±0.014 | 0.521±0.023 | 0.426±0.011 | 0.611±0.009 | 1.6M | 0.02s |
| TOTEM | 0.368±0.195 | 0.526±0.026 | 0.817±0.018 | 0.741±0.009 | 0.868±0.002 | 23.8K | 0.03s |
| FreTS | 0.264±0.020 | 0.294±0.022 | 0.540±0.014 | 0.505±0.014 | 0.661±0.019 | 465.3K | 0.01s |
| Time-LLM | 0.290±0.001 | 0.417±0.030 | 0.567±0.010 | 0.614±0.004 | 0.775±0.004 | 31.1M | 3.81s |
| MOMENT | 0.550±0.145 | 0.757±0.129 | 0.853±0.014 | 0.838±0.013 | 0.924±0.026 | 109.7M | 3.82s |

## B.3. Results on ItalyAir

*Table 9.* Complete MAE results on ItalyAir across all missing patterns. Mean ± std over 5 runs. Ranking: **1st**, 2nd.

| Model | Point-10% | Point-50% | Point-90% | Block-50% | Subseq-50% | #Params↓ | Time↓ |
|---|---|---|---|---|---|---|---|
| *Naive Baselines* | | | | | | | |
| Linear Interpolation | 0.135±N/A | 0.214±N/A | 0.481±N/A | 0.377±N/A | 0.329±N/A | N/A | N/A |
| LOCF | 0.233±N/A | 0.346±N/A | 0.614±N/A | 0.493±N/A | 0.528±N/A | N/A | N/A |
| Median Imputation | 0.518±N/A | 0.533±N/A | 0.533±N/A | 0.589±N/A | 0.549±N/A | N/A | N/A |
| Mean Imputation | 0.574±N/A | 0.588±N/A | 0.598±N/A | 0.625±N/A | 0.612±N/A | N/A | N/A |
| *Deep Learning Methods* | | | | | | | |
| **HELIX (Ours)** | **0.122±0.010** | **0.203±0.009** | **0.463±0.014** | **0.332±0.011** | **0.297±0.015** | 77.3K | 0.18s |
| ImputeFormer | 0.192±0.017 | 0.282±0.015 | 0.535±0.008 | 0.397±0.019 | 0.356±0.009 | 149.1K | 0.09s |
| SAITS | 0.173±0.004 | 0.276±0.008 | 0.478±0.017 | 0.413±0.015 | 0.396±0.006 | 16.6M | 0.04s |
| Nonstationary Trans. | 0.230±0.004 | 0.281±0.004 | 0.496±0.012 | 0.417±0.009 | 0.381±0.008 | 8.4M | 0.02s |
| PatchTST | 0.271±0.018 | 0.340±0.019 | 0.550±0.063 | 0.514±0.015 | 0.496±0.006 | 5.1M | 0.37s |
| iTransformer | 0.220±0.010 | 0.319±0.008 | 0.584±0.003 | 0.504±0.012 | 0.556±0.017 | 18.9M | 0.03s |
| TEFN | 0.142±0.001 | 0.267±0.004 | 0.485±0.004 | 0.421±0.001 | 0.391±0.004 | 551 | 0.03s |
| TimeMixer | 0.580±0.157 | 0.664±0.105 | 0.745±0.015 | 0.797±0.044 | 0.751±0.032 | 9.6K | 0.04s |
| TimeMixer++ | 0.615±0.054 | 0.642±0.018 | 0.698±0.028 | 0.733±0.020 | 0.730±0.018 | 4.2M | 0.15s |
| ModernTCN | 0.348±0.015 | 0.461±0.016 | 0.690±0.024 | 0.614±0.014 | 0.599±0.008 | 58.1K | 0.02s |
| StemGNN | 0.277±0.028 | 0.301±0.008 | 0.468±0.013 | 0.448±0.010 | 0.425±0.018 | 211.6K | 0.03s |
| TOTEM | 0.677±0.108 | 0.698±0.060 | 0.760±0.004 | 0.815±0.021 | 0.757±0.028 | 27.9K | 0.07s |
| FreTS | 0.279±0.018 | 0.320±0.010 | 0.489±0.009 | 0.498±0.017 | 0.447±0.008 | 668.3K | 0.02s |
| Time-LLM | 0.287±0.008 | 0.436±0.016 | 0.567±0.004 | 0.682±0.005 | 0.625±0.004 | 31.1M | 9.44s |
| MOMENT | – | – | – | – | – | N/A | N/A |

## B.4. Results on PeMS

*Table 10.* Complete MAE results on PeMS across all missing patterns. Mean ± std over 5 runs. Ranking: **1st**, 2nd.

| Model | Point-10% | Point-50% | Point-90% | Block-50% | Subseq-50% | #Params↓ | Time↓ |
|---|---|---|---|---|---|---|---|
| *Naive Baselines* | | | | | | | |
| Linear Interpolation | 0.211±N/A | 0.343±N/A | 0.834±N/A | 0.716±N/A | 1.000±N/A | N/A | N/A |
| LOCF | 0.375±N/A | 0.547±N/A | 0.899±N/A | 0.920±N/A | 1.203±N/A | N/A | N/A |
| Median Imputation | 0.778±N/A | 0.777±N/A | 0.779±N/A | 0.856±N/A | 0.886±N/A | N/A | N/A |
| Mean Imputation | 0.798±N/A | 0.799±N/A | 0.800±N/A | 0.829±N/A | 0.849±N/A | N/A | N/A |
| *Deep Learning Methods* | | | | | | | |
| **HELIX (Ours)** | **0.097±0.002** | **0.134±0.001** | **0.256±0.008** | **0.203±0.004** | **0.311±0.092** | 24.0M | 32.32s |
| ImputeFormer | 0.164±0.007 | 0.186±0.007 | 0.278±0.007 | 0.257±0.003 | 0.338±0.011 | 356.2M | 21.84s |
| SAITS | 0.277±0.001 | 0.285±0.001 | 0.317±0.001 | 0.325±0.001 | 0.348±0.001 | 78.2M | 0.08s |
| Nonstationary Trans. | 0.284±0.010 | 0.345±0.002 | 0.712±0.001 | 0.631±0.005 | 0.991±0.006 | 346.3K | 0.06s |
| PatchTST | 0.299±0.004 | 0.310±0.005 | 0.380±0.008 | 0.363±0.006 | 0.408±0.011 | 3.0M | 0.13s |
| iTransformer | 0.181±0.004 | 0.228±0.005 | 0.392±0.008 | 0.422±0.058 | 0.625±0.008 | 1.9M | 0.25s |
| TEFN | 0.284±0.003 | 0.382±0.004 | 0.761±0.001 | 0.693±0.003 | 1.015±0.003 | 1.5M | 0.20s |
| TimeMixer | 0.329±0.004 | 0.328±0.004 | 0.362±0.002 | 0.363±0.002 | 0.388±0.010 | 8.1M | 0.87s |
| TimeMixer++ | – | – | – | – | – | N/A | N/A |
| ModernTCN | 0.486±0.040 | 0.566±0.027 | 0.705±0.014 | 0.624±0.010 | 0.655±0.011 | 1287.0M | 12.88s |
| StemGNN | 0.293±0.001 | 0.300±0.002 | 0.331±0.001 | 0.342±0.002 | 0.365±0.001 | 3.8M | 2.31s |
| TOTEM | 0.282±0.013 | 0.610±0.014 | 0.764±0.011 | 0.738±0.008 | 0.788±0.007 | 443.8K | 0.31s |
| FreTS | 0.342±0.016 | 0.352±0.008 | 0.388±0.004 | 0.408±0.008 | 0.442±0.009 | 1.7M | 0.05s |
| Time-LLM | – | – | – | – | – | N/A | N/A |
| MOMENT | – | – | – | – | – | N/A | N/A |

## B.5. Results on PhysioNet2012

*Table 11.* Complete MAE results on PhysioNet2012 across all missing patterns. Mean ± std over 5 runs. Ranking: **1st**, 2nd.

| Model | Point-10% | #Params↓ | Time↓ |
|---|---|---|---|
| *Naive Baselines* | | | |
| Linear Interpolation | 0.366±N/A | N/A | N/A |
| LOCF | 0.449±N/A | N/A | N/A |
| Median Imputation | 0.690±N/A | N/A | N/A |
| Mean Imputation | 0.708±N/A | N/A | N/A |
| *Deep Learning Methods* | | | |
| **HELIX (Ours)** | **0.215±0.002** | 454.9K | 0.74s |
| ImputeFormer | 0.240±0.009 | 1.4M | 0.89s |
| SAITS | 0.247±0.010 | 44.3M | 0.31s |
| Nonstationary Trans. | 0.316±0.002 | 1.6M | 0.15s |
| PatchTST | 0.282±0.010 | 644.1K | 0.45s |
| iTransformer | 0.373±0.003 | 6.9M | 0.13s |
| TEFN | 0.364±0.001 | 3.1K | 0.10s |
| TimeMixer | 0.429±0.003 | 73.4K | 0.26s |
| TimeMixer++ | 0.416±0.002 | 44.7M | 3.36s |
| ModernTCN | 0.499±0.006 | 850.2K | 0.16s |
| StemGNN | 0.391±0.034 | 6.5M | 0.20s |
| TOTEM | 0.474±0.012 | 86.6K | 0.35s |
| FreTS | 0.317±0.012 | 1.9M | 0.31s |
| Time-LLM | 0.457±0.003 | 31.1M | 146.39s |
| MOMENT | 0.600±0.133 | 35.4M | 2.34s |

HELIX substantially outperforms naive baselines on this challenging medical dataset.

## C. Multi-Metric Consistency

To verify that HELIX's improvements are not specific to MAE, we report average ranks for MAE, MSE, and MRE independently across all 21 settings.

*Table 12.* Multi-metric ranking consistency. Average rank across all 21 settings for MAE, MSE, and MRE (lower is better). HELIX achieves the best average rank on all three metrics. †: incomplete settings.

| Model | MAE Rank↓ | MSE Rank↓ | MRE Rank↓ | Valid |
|---|---|---|---|---|
| **HELIX (Ours)** | **1.00** | **1.38** | **1.00** | 21/21 |
| ImputeFormer | 3.29 | 3.29 | 3.29 | 21/21 |
| SAITS | 3.76 | 3.24 | 3.76 | 21/21 |
| StemGNN | 5.71 | 6.00 | 5.71 | 21/21 |
| Linear Interpolation | 6.67 | 7.57 | 6.67 | 21/21 |
| PatchTST | 7.19 | 6.05 | 7.19 | 21/21 |
| Nonstationary Trans. | 7.29 | 7.71 | 7.33 | 21/21 |
| FreTS | 7.48 | 6.57 | 7.43 | 21/21 |
| iTransformer | 7.95 | 8.10 | 7.90 | 21/21 |
| TEFN | 8.67 | 9.38 | 8.67 | 21/21 |
| Time-LLM | 11.75 | 11.19 | 11.75 | 21/21 |
| TimeMixer | 11.86 | 12.00 | 11.86 | 21/21 |
| LOCF | 12.05 | 13.33 | 12.10 | 21/21 |
| ModernTCN | 12.43 | 12.05 | 12.43 | 21/21 |
| TimeMixer++ | 13.06 | 12.56 | 13.06 | 21/21 |
| TOTEM | 14.38 | 13.90 | 14.38 | 21/21 |
| MOMENT | 16.82 | 16.64 | 16.82 | 21/21 |

HELIX achieves the best average rank on all three metrics. Complete MSE results per dataset are provided below; MRE tables are omitted as MRE ranks are nearly identical to MAE (differing by at most 0.05 in average rank for all models).

*Table 13.* Complete MSE results on BeijingAir across all missing patterns. Mean ± std over 5 runs. Ranking: **1st**, 2nd.

| Model | Point-10% | Point-50% | Point-90% | Block-50% | Subseq-50% |
|---|---|---|---|---|---|
| *Naive Baselines* | | | | | |
| Linear Interpolation | 0.114±N/A | 0.231±N/A | 0.608±N/A | 0.418±N/A | 0.588±N/A |
| LOCF | 0.243±N/A | 0.429±N/A | 0.835±N/A | 0.715±N/A | 0.832±N/A |
| Median Imputation | 1.093±N/A | 1.143±N/A | 1.165±N/A | 1.175±N/A | 1.155±N/A |
| Mean Imputation | 1.035±N/A | 1.078±N/A | 1.099±N/A | 1.114±N/A | 1.102±N/A |
| *Deep Learning Methods* | | | | | |
| **HELIX (Ours)** | 0.151±0.009 | 0.172±0.010 | **0.293±0.005** | 0.204±0.008 | **0.233±0.010** |
| ImputeFormer | **0.095±0.021** | **0.156±0.017** | 0.357±0.005 | **0.203±0.010** | 0.470±0.013 |
| SAITS | 0.129±0.008 | 0.180±0.007 | 0.429±0.017 | 0.217±0.009 | 0.237±0.014 |
| Nonstationary Trans. | 0.227±0.004 | 0.252±0.004 | 0.530±0.002 | 0.365±0.004 | 0.451±0.021 |
| PatchTST | 0.138±0.004 | 0.235±0.005 | 0.380±0.002 | 0.339±0.003 | 0.383±0.003 |
| iTransformer | 0.218±0.004 | 0.255±0.004 | 0.525±0.010 | 0.472±0.080 | 0.914±0.009 |
| TEFN | 0.194±0.015 | 0.252±0.003 | 0.600±0.001 | 0.430±0.003 | 0.517±0.002 |
| TimeMixer | 0.381±0.008 | 0.386±0.007 | 0.434±0.004 | 0.404±0.005 | 0.427±0.003 |
| TimeMixer++ | 0.664±0.112 | 0.829±0.033 | 1.087±0.024 | 1.116±0.015 | 1.122±0.023 |
| ModernTCN | 0.297±0.013 | 0.437±0.024 | 1.094±0.015 | 0.633±0.033 | 0.821±0.024 |
| StemGNN | 0.272±0.009 | 0.292±0.004 | 0.359±0.002 | 0.348±0.007 | 0.393±0.006 |
| TOTEM | 0.461±0.353 | 0.570±0.024 | 1.125±0.006 | 1.008±0.006 | 1.106±0.038 |
| FreTS | 0.179±0.014 | 0.260±0.012 | 0.368±0.015 | 0.320±0.019 | 0.384±0.003 |
| Time-LLM | 0.232±0.005 | 0.272±0.010 | 0.551±0.008 | 0.625±0.082 | 0.893±0.039 |
| MOMENT | 2.981±5.308 | 6.680±8.829 | 1.136±0.031 | 1.549±0.141 | 1.446±0.509 |

*Table 14.* Complete MSE results on ETT_h1 across all missing patterns. Mean ± std over 5 runs. Ranking: **1st**, 2nd.

| Model | Point-10% | Point-50% | Point-90% | Block-50% | Subseq-50% |
|---|---|---|---|---|---|
| *Naive Baselines* | | | | | |
| Linear Interpolation | 0.106±N/A | 0.178±N/A | 1.014±N/A | 0.699±N/A | 1.322±N/A |
| LOCF | 0.277±N/A | 0.491±N/A | 1.337±N/A | 1.317±N/A | 1.619±N/A |
| Median Imputation | 1.044±N/A | 1.022±N/A | 1.052±N/A | 1.079±N/A | 1.337±N/A |
| Mean Imputation | 0.990±N/A | 0.971±N/A | 0.992±N/A | 0.973±N/A | 1.194±N/A |
| *Deep Learning Methods* | | | | | |
| **HELIX (Ours)** | **0.049±0.004** | **0.094±0.009** | **0.387±0.013** | **0.320±0.030** | **0.659±0.029** |
| ImputeFormer | 0.100±0.039 | 0.199±0.047 | 0.505±0.014 | 0.352±0.029 | 0.712±0.037 |
| SAITS | 0.054±0.004 | 0.102±0.007 | 0.388±0.014 | 0.375±0.033 | 0.851±0.055 |
| Nonstationary Trans. | 0.190±0.015 | 0.268±0.009 | 0.574±0.018 | 0.478±0.011 | 0.817±0.027 |
| PatchTST | 0.115±0.009 | 0.152±0.024 | 0.469±0.022 | 0.544±0.025 | 1.024±0.106 |
| iTransformer | 0.153±0.003 | 0.224±0.005 | 0.669±0.015 | 0.461±0.006 | 1.105±0.019 |
| TEFN | 0.231±0.005 | 0.509±0.047 | 0.887±0.056 | 0.718±0.046 | 1.173±0.076 |
| TimeMixer | 0.684±0.428 | 0.909±0.252 | 1.209±0.017 | 1.061±0.086 | 1.396±0.029 |
| TimeMixer++ | 0.215±0.005 | 0.266±0.006 | 0.636±0.006 | 0.499±0.012 | 0.870±0.026 |
| ModernTCN | 0.182±0.012 | 0.362±0.019 | 1.113±0.031 | 0.671±0.020 | 1.143±0.009 |
| StemGNN | 0.155±0.006 | 0.191±0.015 | 0.504±0.047 | 0.358±0.018 | 0.815±0.027 |
| TOTEM | 0.313±0.305 | 0.525±0.048 | 1.167±0.045 | 0.976±0.022 | 1.364±0.006 |
| FreTS | 0.140±0.019 | 0.169±0.022 | 0.533±0.029 | 0.507±0.030 | 0.944±0.056 |
| Time-LLM | 0.168±0.002 | 0.321±0.039 | 0.620±0.020 | 0.681±0.018 | 1.180±0.004 |
| MOMENT | 0.584±0.322 | 0.997±0.322 | 1.257±0.047 | 1.258±0.070 | 1.524±0.097 |

*Table 15.* Complete MSE results on ItalyAir across all missing patterns. Mean ± std over 5 runs. Ranking: **1st**, 2nd.

| Model | Point-10% | Point-50% | Point-90% | Block-50% | Subseq-50% |
|---|---|---|---|---|---|
| *Naive Baselines* | | | | | |
| Linear Interpolation | 0.106±N/A | 0.252±N/A | 0.750±N/A | 0.478±N/A | 0.407±N/A |
| LOCF | 0.220±N/A | 0.511±N/A | 1.110±N/A | 0.736±N/A | 0.900±N/A |
| Median Imputation | 1.045±N/A | 1.116±N/A | 1.151±N/A | 1.192±N/A | 1.151±N/A |
| Mean Imputation | 1.034±N/A | 1.096±N/A | 1.125±N/A | 1.163±N/A | 1.130±N/A |
| *Deep Learning Methods* | | | | | |
| **HELIX (Ours)** | **0.066±0.015** | **0.149±0.012** | 0.669±0.065 | **0.322±0.027** | **0.301±0.024** |
| ImputeFormer | 0.140±0.016 | 0.253±0.026 | 0.897±0.017 | 0.444±0.045 | 0.378±0.036 |
| SAITS | 0.088±0.004 | 0.204±0.009 | 0.604±0.024 | 0.406±0.031 | 0.373±0.018 |
| Nonstationary Trans. | 0.184±0.011 | 0.289±0.008 | 0.862±0.034 | 0.534±0.027 | 0.472±0.011 |
| PatchTST | 0.188±0.021 | 0.302±0.021 | 0.944±0.240 | 0.600±0.020 | 0.615±0.005 |
| iTransformer | 0.173±0.009 | 0.322±0.009 | 0.900±0.009 | 0.652±0.027 | 0.712±0.046 |
| TEFN | 0.115±0.003 | 0.285±0.006 | 0.869±0.008 | 0.550±0.003 | 0.483±0.010 |
| TimeMixer | 0.945±0.361 | 1.208±0.279 | 1.450±0.040 | 1.551±0.126 | 1.431±0.090 |
| TimeMixer++ | 1.002±0.091 | 1.120±0.082 | 1.284±0.095 | 1.390±0.068 | 1.311±0.036 |
| ModernTCN | 0.368±0.029 | 0.605±0.037 | 1.292±0.080 | 0.961±0.041 | 0.940±0.025 |
| StemGNN | 0.215±0.032 | 0.262±0.008 | **0.577±0.046** | 0.478±0.017 | 0.455±0.023 |
| TOTEM | 1.202±0.445 | 1.258±0.257 | 1.487±0.015 | 1.614±0.052 | 1.464±0.055 |
| FreTS | 0.189±0.017 | 0.281±0.012 | 0.670±0.022 | 0.592±0.040 | 0.504±0.023 |
| Time-LLM | 0.282±0.021 | 0.533±0.021 | 0.921±0.007 | 1.173±0.014 | 1.028±0.025 |
| MOMENT | – | – | – | – | – |

*Table 16.* Complete MSE results on PeMS across all missing patterns. Mean ± std over 5 runs. Ranking: **1st**, 2nd.

| Model | Point-10% | Point-50% | Point-90% | Block-50% | Subseq-50% |
|---|---|---|---|---|---|
| *Naive Baselines* | | | | | |
| Linear Interpolation | 0.240±N/A | 0.539±N/A | 1.944±N/A | 1.585±N/A | 2.489±N/A |
| LOCF | 0.600±N/A | 1.094±N/A | 1.939±N/A | 2.266±N/A | 2.963±N/A |
| Median Imputation | 1.492±N/A | 1.476±N/A | 1.487±N/A | 1.790±N/A | 1.877±N/A |
| Mean Imputation | 1.423±N/A | 1.416±N/A | 1.425±N/A | 1.618±N/A | 1.681±N/A |
| *Deep Learning Methods* | | | | | |
| **HELIX (Ours)** | **0.221**±**0.007** | **0.325**±**0.009** | **0.545**±**0.012** | **0.512**±**0.009** | **0.708**±**0.186** |
| ImputeFormer | 0.340±0.013 | 0.410±0.016 | 0.571±0.006 | 0.603±0.005 | 0.734±0.017 |
| SAITS | 0.559±0.002 | 0.581±0.001 | 0.629±0.001 | 0.707±0.001 | 0.750±0.001 |
| Nonstationary Trans. | 0.541±0.014 | 0.617±0.006 | 1.535±0.006 | 1.304±0.016 | 2.143±0.008 |
| PatchTST | 0.542±0.010 | 0.566±0.003 | 0.678±0.017 | 0.734±0.010 | 0.806±0.018 |
| iTransformer | 0.326±0.004 | 0.437±0.005 | 0.730±0.018 | 0.878±0.114 | 1.320±0.041 |
| TEFN | 0.499±0.005 | 0.647±0.012 | 1.640±0.005 | 1.395±0.005 | 2.213±0.008 |
| TimeMixer | 0.630±0.004 | 0.633±0.004 | 0.685±0.005 | 0.762±0.003 | 0.811±0.016 |
| TimeMixer++ | – | – | – | – | – |
| ModernTCN | 0.805±0.045 | 1.040±0.041 | 1.303±0.026 | 1.265±0.017 | 1.354±0.019 |
| StemGNN | 0.589±0.010 | 0.604±0.011 | 0.642±0.002 | 0.744±0.003 | 0.779±0.006 |
| TOTEM | 0.397±0.024 | 1.064±0.023 | 1.403±0.024 | 1.486±0.014 | 1.660±0.009 |
| FreTS | 0.577±0.019 | 0.606±0.012 | 0.686±0.015 | 0.791±0.013 | 0.849±0.015 |
| Time-LLM | – | – | – | – | – |
| MOMENT | – | – | – | – | – |

*Table 17.* Complete MSE results on PhysioNet2012 across all missing patterns. Mean ± std over 5 runs. Ranking: **1st**, 2nd.

| Model | Point-10% |
|---|---|
| *Naive Baselines* | |
| Linear Interpolation | 0.425±N/A |
| LOCF | 0.604±N/A |
| Median Imputation | 1.000±N/A |
| Mean Imputation | 0.967±N/A |
| *Deep Learning Methods* | |
| **HELIX (Ours)** | **0.206**±**0.003** |
| ImputeFormer | 0.229±0.008 |
| SAITS | 0.229±0.009 |
| Nonstationary Trans. | 0.300±0.003 |
| PatchTST | 0.274±0.015 |
| iTransformer | 0.360±0.005 |
| TEFN | 0.376±0.002 |
| TimeMixer | 0.450±0.006 |
| TimeMixer++ | 0.410±0.003 |
| ModernTCN | 0.573±0.016 |
| StemGNN | 0.392±0.048 |
| TOTEM | 0.519±0.017 |
| FreTS | 0.314±0.016 |
| Time-LLM | 0.474±0.006 |
| MOMENT | 0.752±0.269 |

## D. Gated Fusion Analysis

A natural question arises: could the simple averaging in Multi-level Fusion be improved with a learnable gating mechanism? Gating has proven effective in many architectures, from LSTMs (Hochreiter & Schmidhuber, 1997) to recent Gated Attention (Qiu et al., 2025). We investigate this by replacing the averaging operation with a learned fusion gate.

### D.1. Gated Fusion Architecture

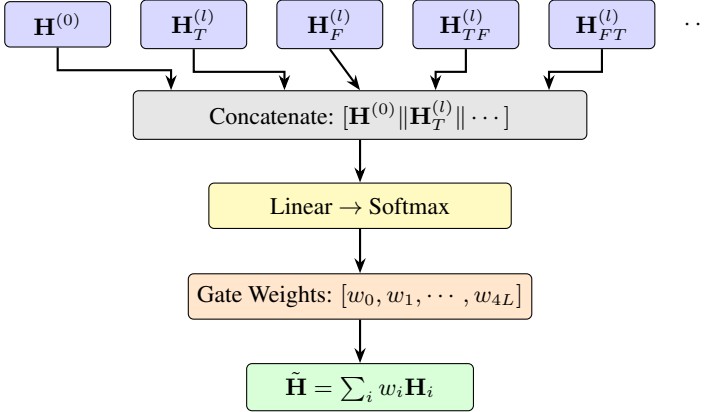

*Figure 6.* Gated Fusion architecture. Input representations are concatenated and passed through a linear layer followed by softmax to produce per-input weights, which are then used for weighted summation.

As shown in Figure 6, the gated fusion mechanism replaces the uniform averaging with learned weights:

$$\mathbf{w} = \text{Softmax}(\mathbf{W}_g[\mathbf{H}^{(0)}\|\mathbf{H}_T^{(1)}\|\cdots\|\mathbf{H}_{FT}^{(L)}]) \tag{15}$$

$$\tilde{\mathbf{H}} = \sum_{i=0}^{4L} w_i \mathbf{H}_i \tag{16}$$

where $\mathbf{W}_g \in \mathbb{R}^{(4L+1)\times(4L+1)\cdot d}$ is a learnable projection matrix.

### D.2. Experimental Results

We conducted the same 25-trial hyperparameter search (refer to Appendix Section G) for the gated variant, using identical search spaces (no additional hyperparameters were introduced for the gate). Table 18 compares the best MAE achieved by each variant.

*Table 18.* Comparison of fusion strategies (Point-10% missing, best MAE from 25-trial search). Gated fusion underperforms simple averaging on 4/5 datasets.

| Dataset | HELIX (Avg) | Gated | w/o Fusion | $\triangle$ Gated |
|---|---|---|---|---|
| BeijingAir | **0.0687** | 0.0925 | 0.0739 | +34.6% |
| ETT-h1 | **0.1209** | 0.1287 | 0.1259 | +6.5% |
| ItalyAir | **0.1001** | 0.1127 | 0.0913 | +12.6% |
| PeMS | 0.0939 | **0.0925** | 0.0942 | $-1.5\%$ |
| PhysioNet2012 | **0.2098** | 0.2133 | 0.2122 | +1.7% |
| *Avg. $\triangle$* | – | – | – | +10.8% |

**Key Finding.** Contrary to expectations, gated fusion *underperforms* simple averaging on 4 out of 5 datasets, with particularly large degradation on BeijingAir (+34.6%). On average, gated fusion increases MAE by 10.8%. Even more surprisingly, gated fusion performs worse than *no fusion at all* on BeijingAir and ItalyAir.

**Interpretation.** We hypothesize that the value of Multi-level Fusion lies in *unconditional information preservation* rather than selective filtering. Different missing patterns and positions require information from different encoding stages: point-wise missing may benefit from fine-grained local features, while block missing requires abstract global patterns. The gating mechanism attempts to learn a single weighting scheme, but this "one-size-fits-all" selection is suboptimal for the heterogeneous information needs across diverse missing scenarios. Simple averaging, by contrast, preserves all information and delegates the selection to the final output projection, which has direct access to the reconstruction target. This finding echoes the success of residual connections (He et al., 2016): sometimes, the simplest aggregation is the most robust.

## E. PhysioNet2012 Feature Embedding Analysis

To validate that Feature Identity Embedding discovers meaningful structure beyond spatial relationships, we analyze learned embeddings on PhysioNet2012: a dataset where features represent physiological measurements with known clinical groupings but no spatial structure.

### E.1. Clinical Feature Groupings

PhysioNet2012 contains 35 ICU monitoring features. We group these according to standard clinical laboratory categories: Blood Pressure (DiasABP, SysABP, MAP, NIDiasABP, NISysABP, NIMAP), Blood Gas (pH, PaO2, PaCO2, HCO3, SaO2, FiO2), Electrolytes (Na, K, Mg), Liver Function (ALT, AST, Bilirubin, ALP, Albumin), Kidney Function (BUN, Creatinine, Urine), Cardiac Markers (TroponinT, TroponinI, HR), Hematology (WBC, HCT, Platelets), and Metabolic (Glucose, Lactate, Cholesterol).

### E.2. Results

Figure 7 compares cosine similarity between feature embeddings for within-group pairs (features from the same clinical category) versus between-group pairs.

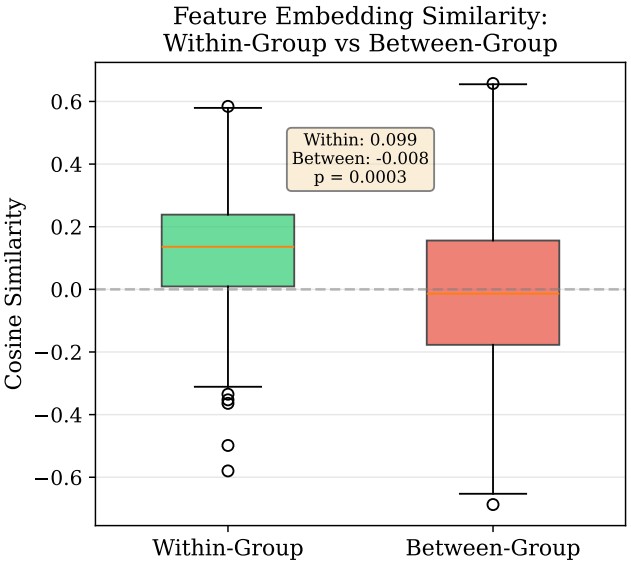

*Figure 7.* Feature Identity Embedding similarity on PhysioNet2012. Within-group pairs (clinically related features) exhibit significantly higher similarity than between-group pairs (mean: 0.099 vs. -0.008, Mann-Whitney U test, $p < 0.001$).

The results confirm that Feature Identity Embedding captures clinically meaningful structure: features within the same functional category develop more similar embeddings than unrelated features. The effect size (difference of 0.107) is smaller than observed on BeijingAir, which we interpret as reflecting the inherent complexity of physiological relationships. Unlike air quality stations where correlation is primarily determined by geographic proximity, ICU measurements exhibit complex interdependencies influenced by underlying pathophysiology, treatment protocols, and temporal dynamics. For instance, liver function markers (ALT, AST) may correlate differently depending on whether the patient has hepatic injury, sepsis, or

cardiac failure. Despite this complexity, Feature Identity Embedding successfully identifies the underlying clinical structure without any supervision.

## F. Ablation Variant Architectures

This section provides detailed architectural specifications for each ablation variant studied in Section 4.3. All variants share the same base architecture except for the specific component being ablated.

### F.1. Notation

*Table 19.* Key architectural parameters.

| Symbol | Description | Default |
|--------|-------------|---------|
| $d_{pe}$ | Temporal positional encoding dimension | 16 |
| $d_f$ | Feature identity embedding dimension | Dataset-specific |
| $d_e$ | Total embedding dimension | $d_{pe} + d_f + 2$ |
| $d$ | Model hidden dimension | 256 |
| $H$ | Number of attention heads | 8 |
| $L$ | Number of encoding layers | 2 |

### F.2. Full HELIX Architecture

The complete HELIX model processes input through:

**1. Time Series Embedding.** Each observation $(t, i)$ is embedded as:

$$\mathbf{e}_{t,i} = [\, x_{t,i} \;;\; \mathrm{PE}(t) \;;\; \mathbf{f}_i \;;\; m_{t,i} \,] \in \mathbb{R}^{d_e} \tag{17}$$

**2. Hybrid Encoding Layer.** Each of $L$ layers performs:

- **Stage 1 (Parallel)**: $\mathbf{H}_T = A_T(\mathbf{H})$, $\mathbf{H}_F = A_F(\mathbf{H})$

- **Stage 2 (Cross-serial)**: $\mathbf{H}_{TF} = A_F(\mathbf{H}_T)$, $\mathbf{H}_{FT} = A_T(\mathbf{H}_F)$

- **Layer fusion**: $\mathbf{H}^{(l)} = \frac{1}{4}(\mathbf{H}_T + \mathbf{H}_F + \mathbf{H}_{TF} + \mathbf{H}_{FT})$

**3. Multi-level Fusion.** Aggregates all intermediate outputs:

$$\tilde{\mathbf{H}} = \frac{1}{1 + 4L}\left(\mathbf{H}^{(0)} + \sum_{l=1}^{L}(\mathbf{H}_T^{(l)} + \mathbf{H}_F^{(l)} + \mathbf{H}_{TF}^{(l)} + \mathbf{H}_{FT}^{(l)})\right) \tag{18}$$

### F.3. Ablation Variant: w/o Feature Identity Embedding

**Modification.** Remove the learnable feature identity embedding $\mathbf{f}_i$ from the embedding layer.

**Embedding change.**

$$\mathbf{e}_{t,i} = [\, x_{t,i} \;;\; \mathrm{PE}(t) \;;\; m_{t,i} \,] \in \mathbb{R}^{d_{pe}+2} \tag{19}$$

**Impact.** Embedding dimension reduces from $d_e = d_{pe} + d_f + 2$ to $d'_e = d_{pe} + 2$. All other components remain unchanged.

### F.4. Ablation Variant: w/o Multi-level Fusion

**Modification.** Remove global multi-level fusion; use only the final layer output.

**Forward pass change.**

$$\tilde{\mathbf{H}} = \text{LayerNorm}(\mathbf{H}^{(L)}) \tag{20}$$

instead of aggregating all intermediate outputs.

**Impact.** Layer-wise fusion within each hybrid encoding layer is *retained*. Only the global aggregation across layers is removed. Intermediate output count: 1 (vs. $1 + 4L$ in full HELIX).

### F.5. Ablation Variant: w/o Hybrid Encoding

**Modification.** Replace the two-stage parallel-serial encoding with pure serial encoding.

**Encoding change.** Each layer performs:

$$\mathbf{H} \leftarrow A_T(\mathbf{H}) \quad \text{(temporal attention)} \tag{21}$$
$$\mathbf{H} \leftarrow A_F(\mathbf{H}) \quad \text{(feature attention)} \tag{22}$$
$$\mathbf{H} \leftarrow A_T(\mathbf{H}) \quad \text{(temporal attention)} \tag{23}$$

**Impact.** No layer-wise fusion (outputs are serially propagated). Intermediate outputs per layer: 3 (vs. 4 in full HELIX). Total intermediate outputs: $1 + 3L$.

### F.6. Ablation Variant: w/o Sinusoidal PE (Learnable PE)

**Modification.** Replace sinusoidal positional encoding with learnable positional embedding of the same dimension.

**Embedding change.**

$$\text{PE}(t) \leftarrow \mathbf{P}_{\text{learn}}[t] \quad \text{where } \mathbf{P}_{\text{learn}} \in \mathbb{R}^{T_{\max} \times d_{pe}} \tag{24}$$

**Impact.** Adds $T_{\max} \times d_{pe}$ learnable parameters. All other components remain unchanged.

### F.7. Summary Comparison

*Table 20.* Architectural comparison of ablation variants.

| Variant | Embed. Dim. | Outputs/Layer | Total Outputs | Layer Fusion |
|---------|-------------|---------------|---------------|--------------|
| Full HELIX | $d_{pe} + d_f + 2$ | 4 | $1 + 4L$ | ✓ |
| w/o FeatID | $d_{pe} + 2$ | 4 | $1 + 4L$ | ✓ |
| w/o Fusion | $d_{pe} + d_f + 2$ | 4 | 1 | ✓ |
| w/o Hybrid | $d_{pe} + d_f + 2$ | 3 | $1 + 3L$ | – |
| w/o Sinusoidal PE | $d_{pe} + d_f + 2$ | 4 | $1 + 4L$ | ✓ |

*Table 21.* Ablation results (MAE) on BeijingAir (24 steps, 132 features). Ranking per row among ablations: **1st**, 2nd.

| Pattern | HELIX (Ours) | w/o Multi-level Fusion | w/o Sinusoidal PE | w/o Hybrid Encoding | w/o Feature Identity Emb. |
|---|---|---|---|---|---|
| **Point-10%** | **0.073±0.004** | 0.073±0.006 | 0.080±0.006 | 0.080±0.003 | 0.113±0.003 |
| **Point-50%** | **0.102±0.005** | 0.104±0.006 | 0.108±0.002 | 0.104±0.004 | 0.144±0.006 |
| **Point-90%** | 0.190±0.005 | 0.194±0.006 | 0.187±0.005 | **0.184±0.002** | 0.321±0.007 |
| **Block-50%** | **0.131±0.005** | 0.147±0.019 | 0.142±0.003 | 0.137±0.004 | 0.223±0.009 |
| **Subseq-50%** | **0.166±0.009** | 0.173±0.014 | 0.173±0.010 | 0.294±0.101 | 0.398±0.016 |

*Table 22.* Ablation results (MAE) on ETT-h1 (48 steps, 7 features). Ranking per row among ablations: **1st**, 2nd.

| Pattern | HELIX (Ours) | w/o Multi-level Fusion | w/o Sinusoidal PE | w/o Hybrid Encoding | w/o Feature Identity Emb. |
|---|---|---|---|---|---|
| **Point-10%** | 0.128±0.005 | **0.123±0.002** | 0.133±0.004 | 0.138±0.014 | 0.193±0.010 |
| **Point-50%** | 0.223±0.014 | 0.192±0.011 | 0.206±0.013 | **0.188±0.007** | 0.241±0.012 |
| **Point-90%** | 0.429±0.011 | **0.411±0.015** | 0.477±0.012 | 0.413±0.017 | 0.446±0.009 |
| **Block-50%** | 0.372±0.015 | **0.352±0.014** | 0.356±0.020 | 0.373±0.005 | 0.395±0.006 |
| **Subseq-50%** | 0.489±0.014 | **0.487±0.009** | 0.546±0.011 | 0.548±0.012 | 0.607±0.021 |

*Table 23.* Ablation results (MAE) on ItalyAir (12 steps, 13 features). Ranking per row among ablations: **1st**, 2nd.

| Pattern | HELIX (Ours) | w/o Multi-level Fusion | w/o Sinusoidal PE | w/o Hybrid Encoding | w/o Feature Identity Emb. |
|---|---|---|---|---|---|
| **Point-10%** | 0.122±0.010 | **0.109±0.013** | 0.139±0.014 | 0.142±0.076 | 0.133±0.005 |
| **Point-50%** | 0.203±0.009 | **0.185±0.013** | 0.233±0.024 | 0.207±0.012 | 0.242±0.010 |
| **Point-90%** | **0.463±0.014** | 0.472±0.014 | 0.486±0.015 | 0.517±0.016 | 0.541±0.006 |
| **Block-50%** | 0.332±0.011 | **0.317±0.028** | 0.373±0.015 | 0.339±0.023 | 0.403±0.017 |
| **Subseq-50%** | 0.297±0.015 | **0.284±0.022** | 0.351±0.025 | 0.308±0.025 | 0.388±0.006 |

*Table 24.* Ablation results (MAE) on PeMS (24 steps, 862 features). Ranking per row among ablations: **1st**, 2nd.

| Pattern | HELIX (Ours) | w/o Multi-level Fusion | w/o Sinusoidal PE | w/o Hybrid Encoding | w/o Feature Identity Emb. |
|---|---|---|---|---|---|
| **Point-10%** | 0.097±0.002 | **0.095±0.003** | 0.096±0.001 | 0.098±0.005 | 0.152±0.003 |
| **Point-50%** | 0.134±0.001 | **0.133±0.002** | 0.133±0.003 | 0.136±0.003 | 0.196±0.002 |
| **Point-90%** | 0.256±0.008 | **0.250±0.006** | 0.266±0.004 | 0.256±0.005 | 0.363±0.004 |
| **Block-50%** | 0.203±0.004 | 0.198±0.002 | **0.195±0.006** | 0.202±0.004 | 0.331±0.004 |
| **Subseq-50%** | **0.311±0.092** | 0.542±0.009 | 0.419±0.116 | 0.355±0.113 | 0.457±0.014 |

*Table 25.* Ablation results (MAE) on PhysioNet2012 (48 steps, 35 features). Ranking per row among ablations: **1st**, 2nd.

| Pattern | HELIX (Ours) | w/o Multi-level Fusion | w/o Sinusoidal PE | w/o Hybrid Encoding | w/o Feature Identity Emb. |
|---|---|---|---|---|---|
| **Point-10%** | 0.215±0.002 | 0.216±0.004 | 0.227±0.005 | **0.207±0.004** | 0.235±0.006 |

# G. Implementation Details

## G.1. Hyperparameter Search Space

*Table 26.* Selected hyperparameters for HELIX across datasets after 25-trial search.

| Dataset | $d_{pe}$ | $d_f$ | $d$ | $H$ | $L$ | Dropout |
|---|---|---|---|---|---|---|
| ETT-h1 | 24 | 12 | 128 | 4 | 2 | 0.0 |
| BeijingAir | 12 | 24 | 256 | 12 | 3 | 0.2 |
| ItalyAir | 6 | 6 | 32 | 2 | 3 | 0.2 |
| PeMS | 6 | 32 | 576 | 6 | 3 | 0.1 |
| PhysioNet2012 | 16 | 16 | 96 | 8 | 2 | 0.1 |

*Table 27.* HELIX Hyperparameter Search Space

| Parameter | ETT-h1 | BeijingAir | ItalyAir | PeMS | PhysioNet2012 |
|---|---|---|---|---|---|
| *Dataset Properties* | | | | | |
| n_steps | 48 | 24 | 12 | 24 | 48 |
| n_features | 7 | 132 | 13 | 862 | 35 |
| *Model Architecture* | | | | | |
| d_model | {96, 128, 192, 256} | {192, 256, 384} | {32, 40, 64} | {384, 512, 576, 768} | {64, 96, 128} |
| dropout | {0, 0.1, 0.2, 0.3} | {0, 0.1, 0.2} | {0, 0.1, 0.2} | {0, 0.1, 0.2} | {0, 0.1, 0.2} |
| d_feature_embed | {6, 12, 24} | {16, 24, 32} | {4, 6, 8} | {32, 64, 128} | {8, 10, 16} |
| n_heads | {4, 6, 8} | {4, 8, 12} | {2, 4, 8} | {6, 8, 12} | {2, 4, 8} |
| n_layers | {1, 2, 3} | {1, 2, 3} | {1, 2, 3} | {1, 2, 3} | {1, 2, 3} |
| d_pe | {12, 24, 48} | {6, 12, 24} | {3, 4, 6} | {6, 12, 24} | {8, 12, 16} |
| *Training Configuration* | | | | | |
| epochs | 1000 | 1000 | 1000 | 1000 | 1000 |
| patience | 10 | 10 | 10 | 10 | 10 |
| batch_size | {8, 16, 32} | {4, 8, 16} | {8, 16, 32} | {1, 2} | {8, 16, 32} |
| lr | [1e-4, 0.01] (log) | [1e-4, 0.01] (log) | [1e-4, 0.01] (log) | [1e-4, 0.005] (log) | [1e-4, 0.01] (log) |
| *Loss Weights* | | | | | |
| ORT_weight | – | 1.0 | 1.0 | – | 1.0 |
| MIT_weight | – | 1.0 | 1.0 | – | 1.0 |

*Note: The ablation variants (w/o Sinusoidal PE, w/o Feature Identity Embedding, w/o Hybrid Encoding, w/o Multi-level Fusion) use the same search space.*

*Table 28.* TEFN Hyperparameter Search Space

| Parameter | ETT-h1 | BeijingAir | ItalyAir | PeMS | PhysioNet2012 |
|---|---|---|---|---|---|
| *Dataset Properties* | | | | | |
| n_steps | 48 | 24 | 12 | 24 | 48 |
| n_features | 7 | 132 | 13 | 862 | 35 |
| *Model Architecture* | | | | | |
| apply_nonstationary_norm | {T, F} | {T, F} | {T, F} | {T, F} | {T, F} |
| n_fod | {1, 2, 3} | {1, 2, 3} | {1, 2, 3} | {1, 2, 3} | {1, 2, 3} |
| *Training Configuration* | | | | | |
| epochs | 1000 | 1000 | 1000 | 1000 | 1000 |
| patience | 10 | 10 | 10 | 10 | 10 |
| batch_size | {8, 16, 32} | {4, 8, 16} | {8, 16, 32} | {1, 2} | {8, 16, 32} |
| lr | [1e-4, 0.01] (log) | [1e-4, 0.01] (log) | [1e-4, 0.01] (log) | [1e-4, 0.005] (log) | [1e-4, 0.01] (log) |
| *Loss Weights* | | | | | |
| ORT_weight | – | 1.0 | 1.0 | – | 1.0 |
| MIT_weight | – | 1.0 | 1.0 | – | 1.0 |

*Table 29.* TimeMixer Hyperparameter Search Space

| Parameter | ETT-h1 | BeijingAir | ItalyAir | PeMS | PhysioNet2012 |
|---|---|---|---|---|---|
| *Dataset Properties* | | | | | |
| n_steps | 48 | 24 | 12 | 24 | 48 |
| n_features | 7 | 132 | 13 | 862 | 35 |
| *Model Architecture* | | | | | |
| apply_nonstationary_norm | False | False | False | False | False |
| channel_independence | {T, F} | {T, F} | {T, F} | {T, F} | {T, F} |
| d_ffn | {64, 128, 256} | {128, 256, 384} | {48, 64, 96} | {256, 512, 1024} | {64, 128, 256} |
| d_model | {32, 64, 128} | {64, 128, 192} | {24, 32, 48} | {128, 256, 512} | {32, 64, 128} |
| decomp_method | "moving_avg" | "moving_avg" | "moving_avg" | "moving_avg" | "moving_avg" |
| downsampling_layers | {1, 2} | {1, 2} | {1, 2} | {1, 2} | {1, 2} |
| downsampling_window | {2, 4} | {2, 3, 4} | {2, 3} | {2, 4} | {2, 3, 4} |
| dropout | {0, 0.1, 0.2} | {0, 0.1, 0.2} | {0, 0.1, 0.2} | {0, 0.1, 0.2} | {0, 0.1, 0.2} |
| moving_avg | {5, 13, 25} | {3, 5, 7} | {3, 5} | {5, 13, 25} | {3, 5, 7} |
| n_layers | {1, 2, 3} | {1, 2, 3} | {1, 2, 3} | {1, 2, 3} | {1, 2, 3} |
| top_k | {3, 5, 7} | {3, 5, 7} | {2, 3, 5} | {3, 5, 7} | {3, 5, 7} |
| *Training Configuration* | | | | | |
| epochs | 1000 | 1000 | 1000 | 1000 | 1000 |
| patience | 10 | 10 | 10 | 10 | 10 |
| batch_size | {8, 16, 32} | {4, 8, 16} | {8, 16, 32} | {1, 2} | {8, 16, 32} |
| lr | [1e-4, 0.01] (log) | [1e-4, 0.01] (log) | [1e-4, 0.01] (log) | [1e-4, 0.005] (log) | [1e-4, 0.01] (log) |

*Table 30.* ModernTCN Hyperparameter Search Space

| Parameter | ETT-h1 | BeijingAir | ItalyAir | PeMS | PhysioNet2012 |
|---|---|---|---|---|---|
| *Dataset Properties* | | | | | |
| n_steps | 48 | 24 | 12 | 24 | 48 |
| n_features | 7 | 132 | 13 | 862 | 35 |
| *Model Architecture* | | | | | |
| apply_nonstationary_norm | False | False | False | False | False |
| backbone_dropout | {0, 0.1, 0.2} | {0, 0.1, 0.2} | {0, 0.1, 0.2} | {0, 0.1, 0.2} | {0, 0.1, 0.2} |
| dims | {[32, 32], [64, 64], [32, 64]} | {[48, 48], [64, 64], [48, 64]} | {[24], [32], [24, 32]} | {[64, 64], [128, 128], [64, 128]} | {[24, 24], [32, 32], [24, 32]} |
| downsampling_ratio | {2, 4} | {2, 4} | {2, 3} | {2, 4} | {2, 4} |
| ffn_ratio | {2, 4} | {2, 4} | {2, 4} | {2, 4} | {2, 4} |
| head_dropout | {0, 0.1, 0.2} | {0, 0.1, 0.2} | {0, 0.1, 0.2} | {0, 0.1, 0.2} | {0, 0.1, 0.2} |
| individual | False | False | False | False | False |
| large_size | [7, 7] | {[7, 7], [5, 5]} | {[5], [5, 5]} | [7, 7] | {[7, 7], [5, 5]} |
| num_blocks | [1, 1] | {[1, 1], [1, 2]} | {[1], [1, 1]} | [1, 1] | {[1, 1], [1, 2]} |
| patch_size | {4, 6, 8} | {4, 6, 8} | {3, 4, 6} | {4, 6, 8} | {6, 8, 12} |
| patch_stride | {4, 6, 8} | {4, 6, 8} | {3, 4, 6} | {4, 6, 8} | {6, 8, 12} |
| small_kernel_merged | False | False | False | False | False |
| small_size | [3, 3] | [3, 3] | {[3], [3, 3]} | [3, 3] | [3, 3] |
| use_multi_scale | False | False | False | False | False |
| *Training Configuration* | | | | | |
| epochs | 1000 | 1000 | 1000 | 1000 | 1000 |
| patience | 10 | 10 | 10 | 10 | 10 |
| batch_size | {8, 16, 32} | {4, 8, 16} | {8, 16, 32} | {1, 2} | {8, 16, 32} |
| lr | [1e-4, 0.01] (log) | [1e-4, 0.01] (log) | [1e-4, 0.01] (log) | [1e-4, 0.005] (log) | [1e-4, 0.01] (log) |

*Table 31.* ImputeFormer Hyperparameter Search Space

| Parameter | ETT-h1 | BeijingAir | ItalyAir | PeMS | PhysioNet2012 |
|---|---|---|---|---|---|
| *Dataset Properties* | | | | | |
| n_steps | 48 | 24 | 12 | 24 | 48 |
| n_features | 7 | 132 | 13 | 862 | 35 |
| *Model Architecture* | | | | | |
| d_ffn | {32, 64, 128} | {96, 128, 192} | {32, 48, 64} | {128, 256, 512} | {64, 96, 128} |
| d_input_embed | {16, 32, 64} | {48, 64, 96} | {16, 24, 32} | {64, 128, 256} | {32, 48, 64} |
| d_learnable_embed | {16, 32, 64} | {48, 64, 96} | {16, 24, 32} | {64, 128, 256} | {32, 48, 64} |
| d_proj | {16, 32, 64} | {48, 64, 96} | {16, 24, 32} | {64, 128, 256} | {32, 48, 64} |
| dropout | {0, 0.1, 0.2} | {0, 0.1, 0.2} | {0, 0.1, 0.2} | {0, 0.1, 0.2} | {0, 0.1, 0.2} |
| input_dim | – | 1 | 1 | – | 1 |
| n_layers | {1, 2, 3} | {1, 2, 3} | {1, 2, 3} | {1, 2, 3} | {1, 2, 3} |
| n_temporal_heads | {2, 4, 8} | {2, 4, 8} | {2, 4, 8} | {4, 8, 16} | {2, 4, 8} |
| output_dim | – | 1 | 1 | – | 1 |
| *Training Configuration* | | | | | |
| epochs | 1000 | 1000 | 1000 | 1000 | 1000 |
| patience | 10 | 10 | 10 | 10 | 10 |
| batch_size | {8, 16, 32} | {4, 8, 16} | {8, 16, 32} | {1, 2} | {8, 16, 32} |
| lr | [1e-4, 0.01] (log) | [1e-4, 0.01] (log) | [1e-4, 0.01] (log) | [1e-4, 0.005] (log) | [1e-4, 0.01] (log) |
| *Loss Weights* | | | | | |
| ORT_weight | – | 1.0 | 1.0 | – | 1.0 |
| MIT_weight | – | 1.0 | 1.0 | – | 1.0 |

*Table 32.* TOTEM Hyperparameter Search Space

| Parameter | ETT-h1 | BeijingAir | ItalyAir | PeMS | PhysioNet2012 |
|---|---|---|---|---|---|
| *Dataset Properties* | | | | | |
| n_steps | 48 | 24 | 12 | 24 | 48 |
| n_features | 7 | 132 | 13 | 862 | 35 |
| *Model Architecture* | | | | | |
| commitment_cost | {0.1, 0.25, 0.5} | {0.1, 0.25, 0.5} | {0.1, 0.25, 0.5} | {0.1, 0.25, 0.5} | {0.1, 0.25, 0.5} |
| compression_factor | {2, 4, 8} | {2, 3, 4} | {2, 3, 4} | {2, 4, 8} | {2, 3, 4} |
| d_block_hidden | {16, 32, 64} | {48, 64, 96} | {16, 24, 32} | {64, 128, 256} | {32, 48, 64} |
| d_embedding | {16, 32, 64} | {48, 64, 96} | {16, 24, 32} | {64, 128, 256} | {32, 48, 64} |
| d_residual_hidden | {8, 16, 32} | {24, 32, 48} | {8, 12, 16} | {32, 64, 128} | {16, 24, 32} |
| n_embeddings | {128, 256, 512} | {256, 512, 768} | {64, 128, 256} | {256, 512, 1024} | {128, 256, 512} |
| n_residual_layers | {1, 2, 3} | {1, 2, 3} | {1, 2, 3} | {1, 2, 3} | {1, 2, 3} |
| *Training Configuration* | | | | | |
| epochs | 1000 | 1000 | 1000 | 1000 | 1000 |
| patience | 10 | 10 | 10 | 10 | 10 |
| batch_size | {8, 16, 32} | {4, 8, 16} | {8, 16, 32} | {1, 2} | {8, 16, 32} |
| lr | [1e-4, 0.01] (log) | [1e-4, 0.01] (log) | [5e-5, 0.001] (log) | [1e-4, 0.005] (log) | [1e-4, 0.01] (log) |

*Table 33.* TimeMixer++ Hyperparameter Search Space

| Parameter | ETT-h1 | BeijingAir | ItalyAir | PeMS | PhysioNet2012 |
|---|---|---|---|---|---|
| *Dataset Properties* | | | | | |
| n_steps | 48 | 24 | 12 | – | 48 |
| n_features | 7 | 132 | 13 | – | 35 |
| *Model Architecture* | | | | | |
| apply_nonstationary_norm | False | False | False | – | False |
| channel_independence | {T, F} | {T, F} | {T, F} | – | {T, F} |
| channel_mixing | {T, F} | {T, F} | {T, F} | – | {T, F} |
| d_ffn | {64, 128, 256} | {128, 256, 384} | {48, 64, 96} | – | {64, 128, 256} |
| d_model | {32, 64, 128} | {64, 128, 192} | {24, 32, 48} | – | {32, 64, 128} |
| downsampling_layers | {1, 2} | {1, 2} | {1, 2} | – | {1, 2} |
| downsampling_window | {2, 4} | {2, 3, 4} | {2, 3} | – | {2, 3, 4} |
| dropout | {0, 0.1, 0.2} | {0, 0.1, 0.2} | {0, 0.1, 0.2} | – | {0, 0.1, 0.2} |
| n_heads | {2, 4, 8} | {2, 4, 8} | {2, 4, 8} | – | {2, 4, 8} |
| n_kernels | {4, 6, 8} | {4, 6, 8} | {4, 6, 8} | – | {4, 6, 8} |
| n_layers | {1, 2, 3} | {1, 2, 3} | {1, 2, 3} | – | {1, 2, 3} |
| top_k | {3, 5, 7} | {3, 5, 7} | {2, 3, 5} | – | {2, 3, 5} |
| *Training Configuration* | | | | | |
| epochs | 1000 | 1000 | 1000 | – | 1000 |
| patience | 10 | 10 | 10 | – | 10 |
| batch_size | {8, 16, 32} | {4, 8, 16} | {8, 16, 32} | – | {8, 16, 32} |
| lr | [1e-4, 0.01] (log) | [1e-4, 0.01] (log) | [1e-4, 0.01] (log) | – | [1e-4, 0.01] (log) |

*Note: TimeMixer++ was not evaluated on PeMS due to excessive memory requirements.*

*Table 34.* TimeLLM Hyperparameter Search Space

| Parameter | ETT-h1 | BeijingAir | ItalyAir | PeMS | PhysioNet2012 |
|---|---|---|---|---|---|
| *Dataset Properties* | | | | | |
| n_steps | 48 | – | – | – | – |
| n_features | 7 | – | – | – | – |
| *Model Architecture* | | | | | |
| d_ffn | {32, 64, 128} | – | – | – | – |
| d_llm | 768 | – | – | – | – |
| d_model | {16, 32, 64} | – | – | – | – |
| domain_prompt_content | (see note) | – | – | – | – |
| dropout | {0, 0.1, 0.2} | – | – | – | – |
| llm_model_type | "BERT" | – | – | – | – |
| n_heads | {2, 4, 8} | – | – | – | – |
| n_layers | {1, 2, 3} | – | – | – | – |
| patch_size | {8, 12, 16} | – | – | – | – |
| patch_stride | {8, 12, 16} | – | – | – | – |
| *Training Configuration* | | | | | |
| epochs | 1000 | – | – | – | – |
| patience | 10 | – | – | – | – |
| batch_size | {8, 16, 32} | – | – | – | – |
| lr | [5e-5, 0.001] (log) | – | – | – | – |

*Note: Time-LLM was only evaluated on ETT-h1 due to its computational requirements. The domain_prompt_content was set to "Electricity transformer temperature time series data".*

*Table 35.* MOMENT Hyperparameter Search Space

| Parameter | ETT-h1 | BeijingAir | ItalyAir | PeMS | PhysioNet2012 |
|---|---|---|---|---|---|
| *Dataset Properties* | | | | | |
| n_steps | 48 | 24 | – | – | 48 |
| n_features | 7 | 132 | – | – | 35 |
| *Model Architecture* | | | | | |
| add_positional_embedding | True | True | – | – | True |
| d_ffn | {1024, 2048, 4096} | {1024, 2048, 4096} | – | – | {512, 1024, 2048} |
| d_model | 768 | 768 | – | – | 512 |
| dropout | {0, 0.1, 0.2} | {0, 0.1, 0.2} | – | – | {0, 0.1, 0.2} |
| finetuning_mode | {"linear-probing", "end-to-end"} | {"linear-probing", "end-to-end"} | – | – | {"linear-probing", "end-to-end"} |
| head_dropout | {0, 0.1, 0.2} | {0, 0.1, 0.2} | – | – | {0, 0.1, 0.2} |
| mask_ratio | {0.1, 0.3, 0.5} | {0.1, 0.3, 0.5} | – | – | {0.1, 0.3, 0.5} |
| n_layers | {2, 4, 6} | {2, 4, 6} | – | – | {2, 4} |
| orth_gain | 1.41 | 1.41 | – | – | 1.41 |
| patch_size | {8, 12, 16} | {6, 8, 12} | – | – | {12, 24} |
| patch_stride | {8, 12, 16} | {6, 8, 12} | – | – | {12, 24} |
| revin_affine | True | True | – | – | True |
| transformer_backbone | "t5-base" | "t5-base" | – | – | "t5-small" |
| transformer_type | "encoder_only" | "encoder_only" | – | – | "encoder_only" |
| value_embedding_bias | True | True | – | – | True |
| *Training Configuration* | | | | | |
| epochs | 1000 | 1000 | – | – | 100 |
| patience | 10 | 10 | – | – | 5 |
| batch_size | {8, 16, 32} | {2, 4, 8} | – | – | {8, 16, 32} |
| lr | [5e-5, 0.001] (log) | [5e-5, 0.001] (log) | – | – | [5e-4, 0.005] (log) |

*Note: MOMENT was not evaluated on ItalyAir and PeMS due to sequence length constraints (requires minimum patch count).*

## G.2. Computational Resources

All experiments were conducted on NVIDIA A100 GPUs. Total computational cost: approximately 1,500 GPU hours for all experiments including hyperparameter search.

