# OpenReview forum: "HELIX: Hybrid Encoding with Learnable Identity and Cross-dimensional Synthesis for Time Series Imputation"
_ICML.cc/2026/Conference — ICML 2026 spotlight_

### Official Review · Reviewer_YJBM · 2026-03-07

**Soundness:** 3
**Presentation:** 3
**Significance:** 3
**Originality:** 3
**Overall Recommendation:** 4
**Confidence:** 3

**Summary:**

This paper proposes HELIX, a model for multivariate time-series imputation. The core idea is to assign each feature a learned feature identity embedding, which is combined with the observed value, temporal positional encoding, and missingness mask to form the input representation. On top of this, the model uses a hybrid encoder that mixes temporal attention and cross-feature attention in a two-stage design, aiming to capture both within-feature temporal dependencies and correlations across variables.

The main motivation is that, even when the value of a feature is missing, its identity embedding can still provide a stable signal for modeling feature relationships. This is intended to make cross-feature reasoning more robust under missing data. The paper further introduces a multi-branch encoding scheme and a fusion mechanism to aggregate information from different temporal-feature interaction paths.

Experimentally, the paper reports strong results on several public benchmarks, showing that HELIX consistently outperforms all compared baselines under different missingness settings. These results suggest that the proposed feature-identity-based design is effective for time-series imputation.

However, there are also limitations. In particular, the learned feature identity embeddings are dataset-specific and tied to a fixed feature schema, which makes their generalization to other datasets or changing feature sets unclear. As a result, while the method is empirically strong, its transferability beyond the evaluated benchmarks may be limited

**Compliance With Llm Reviewing Policy:**

Affirmed.

**Final Justification:**

The authors address all my concerns. I would like to raise my score to weak accept.

**Key Questions For Authors:**

My first question concerns the loss function. I may be misunderstanding it, but the ratio $\frac{M_{ORT}}{O}$ does not seem to be clearly specified. Since $L_{ORT}$ and $L_{MIT}$ are given the same weight, I wonder whether the authors experimented with different weighting schemes and, if so, how sensitive the performance is to this choice. More generally, because FeatID is optimized only indirectly through the imputation objective, without any dedicated loss term, it is unclear how the model ensures that these embeddings actually capture the intended notion of feature identity rather than simply learning dataset-specific shortcuts.

A related concern is that feature identity is learned solely through the final imputation loss, without any explicit supervision or regularization that would encourage it to represent true feature semantics. As a result, it remains unclear whether these embeddings reflect intrinsic feature properties, as claimed, or merely absorb dataset-specific correlation patterns that happen to improve imputation performance. Although the Emergent Structure Discovery analysis suggests that FeatID captures some meaningful structural information, it is still unclear whether the learned embeddings also contain other types of information that are actually responsible for the observed performance gains.

**Limitations:**

Yes.

Meanwhile, the reason why feature identity improves the performance is not convincing enough, although the Emergent Structure Discovery somehow proves that it learns the structure information.

**Strengths And Weaknesses:**

Here is a polished version:

Strengths:

The paper is well written and easy to follow.

The proposed idea is intuitive and reasonably motivated.

The experimental evaluation is thorough and shows that the proposed model consistently outperforms the baselines.

The analysis, interpretation, and ablation studies provide useful evidence that the proposed components can work effectively together.

Weaknesses:

Although the idea is straightforward and appealing, the training objective does not explicitly ensure that the learned feature identity embeddings actually capture the intended concept of feature identity. Without additional supervision or constraints, it is unclear whether these embeddings truly reflect intrinsic feature properties or simply absorb dataset-specific correlation patterns. As a result, I am not fully convinced that the paper’s proposed explanation is the main reason for the performance improvement.

The Hybrid Encoding design appears somewhat complicated, while the ablation results suggest that its gains are not consistently significant. This makes it unclear whether the additional architectural complexity is fully justified.

---

> ### Author Rebuttal · Authors · 2026-03-30
>
> We appreciate the reviewer's acknowledgment of the clarity of presentation, motivational aspects, extensive experimental evaluation, and good quality of the ablation studies and interpretations provided in this paper. We hope that our responses to these issues clearly address the reviewer's remaining concerns and welcome any additional feedback. We would be honored by your positive recommendation.
>
> ---
>
> **Q1**
> We thank the reviewer for your thoughtful comments. The equal weighting applied between the ORT and MIT loss functions was inspired by SAITS (Du et al., 2023), as well as being the default setting in the PyPOTS architecture framework shared by all baseline models evaluated in our proposed benchmark. Accordingly, we did not conduct weight sensitivity experiments, as doing so would break comparability with all other evaluated methods and fall outside the scope of our architectural contributions.
>
> We agree that indirect optimization does not provide conclusive evidence that the use of FeatID accurately represents the conceptualization of feature identity. However, there are three aspects that are difficult to reconcile with a pure shortcut model:
>
> (1) Cross-Dataset Consistency in Ablation Results. Removing Feature ID (Tab 4 and 15–19) significantly impacted performance on each of the 5 datasets and all 21 test scenarios with significance p<0.001(Tab 3). These results showcase the importance of FeatID in HELIX and indicate that the learned feature identity embedding is the main reason for the performance improvement.
>
> (2) Degradation Pattern w/o FeatID. The most severe degradation occurs under Subseq missing patterns (Tab 4, 0.166 to 0.398 +140%). In this regime, entire temporal segments of a feature are absent. The only available signal for cross-feature reasoning is the identity of each feature itself, making FeatID's removal most damaging.
>
> (3) Layer-wise attention dynamics (Fig 3). As layer depth increases from one layer to another, the similarity between feature attention induced by the use of FeatID and real-world geometric structure also increases (from 0.589 to 0.712), indicating an active relational refinement process instead of a passive propagation of a static input-level shortcut.
>
>
>
> We do agree that what any learned representation “actually captures” may never be formally proved, which is a common challenge faced by positional encoding, token embedding, and representation learning broadly. In addition to addressing this issue in the original version of this paper, it has been further elaborated upon within the revisions made to the paper.
>
> ---
>
> **Q2**
> Regarding hybrid encoding, as described in the ablation study (Tab 15-19), we found that “not consistently statistically significantly better” best characterizes the performance of w/o Multi-Level Fusion compared to HELIX.
>
> Ablation w/o Hybrid Encoding demonstrated significant reductions in performance under highly structurally challenging missing patterns (e.g., Tab 15 +77% MAE on Beijing Air Subseq-50%: 0.166 → 0.294, Tab 17 +12% on ETT-h1 Subseq-50%: Tab 18 0.489 → 0.548, +14% on PeMS Subseq-50%: 0.311 → 0.355). Although there are some slight improvements in performance for w/o Hybrid Encoding over HELIX in high rate point missingness settings (less than 2% in BeijingAir Point-90% setting), these differences fell squarely within run-to-run variance. This represents proper operation of the design and not over-engineering.
>
> Concerning Multi-Level Fusion: Performance gains from removing Fusion in isolated settings reflect an evaluation framing issue, not a design flaw. Tab 15 shows a clear asymmetry: when Fusion underperforms, the gap is modest (∼5%); when it helps, the gain is substantial (∼40%). Since missing data in multivariate time series will arise at random times and in mixed patterns, rather than optimizing each component separately for one type of missing pattern, a robust model needs to operate successfully under all possible types of missing patterns.
>
> Overall, across all 21 test scenarios, HELIX demonstrated the greatest total performance. Furthermore, both hybrid encoding and multi-level fusion contributed positively to HELIX's overall robustness. Additionally, we consider it reasonable to sacrifice minor performance losses in easier testing conditions for substantial gains in difficult testing conditions.

---

> > ### Author Rebuttal · Reviewer_YJBM · 2026-04-04
> >
> > The authors address all my concerns. I would like to raise my score to weak accept.

---

### Official Review · Reviewer_8VMy · 2026-03-09

**Soundness:** 3
**Presentation:** 4
**Significance:** 3
**Originality:** 3
**Overall Recommendation:** 5
**Confidence:** 4

**Summary:**

This paper proposes HELIX, a hybrid encoding model for multivariate time series imputation. It introduces Feature Identity Embedding (FeatID) as persistent semantic anchors and a double-helix hybrid encoding architecture that interleaves temporal and cross-feature attention. Unlike graph-based or attention-based baselines, HELIX learns feature dependencies end-to-end without predefined topologies, achieving state-of-the-art performance across 5 datasets and 21 experimental settings.

**Compliance With Llm Reviewing Policy:**

Affirmed.

**Key Questions For Authors:**

The evaluation focuses on Point-X% (X=10,50,90), Block-50%, and Subseq-50% patterns, but real-world applications often involve more extreme missing patterns (e.g., Point-95% or Block-90%). How would HELIX perform under these more severe missing patterns, and what are the practical implications for applications with very high missing rates?
The paper introduces "Feature Identity Embedding" as a "persistent semantic anchor," but the theoretical justification for why this particular design is superior remains somewhat qualitative. Could you provide more rigorous theoretical analysis or a formal proof of why this approach leads to better cross-feature attention?

**Limitations:**

yes

**Strengths And Weaknesses:**

Strengths:
• Model novelty: FeatID provides persistent feature-specific semantics, and the double-helix design enables coordinated temporal-feature information flow, addressing limitations of existing methods.
• Strong empirical results: Outperforms 16 baselines (including attention-based, GNN, and foundation models) on diverse datasets, with significant improvements in severe missing scenarios.
• Rigorous ablation and analysis: Ablation studies validate the necessity of FeatID and hybrid encoding; mechanistic analysis shows alignment with latent physical/clinical structure.
Weaknesses:
• Limited cross-dataset transfer: FeatID is learned per dataset, with no analysis on transferability to new datasets.
• Lack of uncertainty quantification: No assessment of imputation uncertainty, limiting utility in high-stakes applications.
• Missing generative model comparison: Omits diffusion-based baselines without detailed trade-off analysis.
• Evaluation cross-dataset transferability by pre-training FeatID on large datasets and fine-tuning on small ones, if possible.
• Test scalability on ultra-high-dimensional time series (F>10³) and optimize if needed.

---

> ### Author Rebuttal · Authors · 2026-03-30
>
> We would like to express our gratitude for the reviewer’s recognition of the novelty, experimental strength, and mechanism of FeatID. We are delighted that our work resonated with your assessment, and thank you for your strong initial support! We respond to each of your comments below, and hope the clarifications warrant your highest recommendation.
>
> **W1**
> Feature Identity Embeddings are not equivalent to NLP token embeddings which have a shared vocabulary across all texts. Due to differing physical units, measurement scales, and statistical distributions defining feature spaces across datasets, transferring embeddings across datasets is unreliable even for identically named features. We leave a systematic investigation of cross-dataset transferability of Feature Identity Embeddings as an interesting direction for future work.
>
> **W2**
> HELIX targets fast, deterministic point prediction, a fundamentally different paradigm from generative models (CSDI, GP-VAE) that output full conditional distributions at the cost of higher latency and stochastic point-accuracy degradation. We note this trade-off in our Impact Statement. Lightweight UQ (conformal prediction, MC dropout) remains planned future work.
>
> **W3**
> We did not ignore diffusion models. Two lines of evidence support our design choice:
>
> * **TSI-Bench:** reveals that all three generative baselines suffer from substantially higher error than leading predictive models. CSDI shows high cross-dataset variance, MAE of 0.102 on BeijingAir but 0.539±0.418 on ItalyAir and 1.483±0.459 on Electricity, variance ~10× larger than HELIX and most predictive baselines. US-GAN and GP-VAE perform even worse.
> * **Direct comparison on BeijingAir Point-10%:** HELIX (0.073±0.004) outperforms CSDI (0.102±0.010) by **28.4%**, with 171x faster inference speed (2.44s from Tab 2 in HELIX vs 417.52s from Tab 11 in tsi-bench).
>
> Full evaluation across 5 datasets x 5 patterns x 25 HPO trials was computationally impractical due to the high computational expense required to train CSDI.
>
> **Q1**
> Point-90% (90% missing) is already included. Two key observations from Tab 7:
>
> | Pattern    | HELIX       | Best Baseline      | Gap   |
> | ---------- | ----------- | -------------------- | ----- |
> | Point-50%  | 0.102±0.005 | 0.122 (ImputeFormer) | 16.4% |
> | Block-50%  | 0.131±0.005 | 0.158 (ImputeFormer) | 17.1% |
> | Point-90%  | 0.190±0.005 | 0.245 (ImputeFormer) | 22.4% |
> | Subseq-50% | 0.166±0.009 | 0.217 (SAITS)        | 23.5% |
>
> 1. The gap at Point-90% (22.4%) is significantly **greater** than that at Point-50% (16.4%), suggesting that HELIX’s advantage does not decrease with increasing levels of missingness.
> 2. The greatest gap occurs at Subseq-50%, the most typical type of failure mode (loss of communication), again illustrating HELIX’s strong robustness in real-world applications.
>
> **Q2**
> **Decomposing Attention Scores.** Let $\\mathbf{e}\_{t,i} = [\\tilde{x}\_{t,i}; \\text{PE}(t); \\mathbf{f}\_{i}; m\_{t,i}]$. The cross-feature attention score decomposes as:
>
> $s\_{ij}^{(t)} = \\underbrace{\\mathbf{f}\_{i}^\\top \\mathbf{A}'\_{ff} \\mathbf{f}\_{j}}\_{ \\text{(I) Static identity bias}} + \\underbrace{\\text{identity} \\times \\text{value interactions}}\_{\\text{(II)}} + \\underbrace{\\text{identity} \\times \\text{mask interactions}}\_{\\text{(III)}} + \\underbrace{\\text{value/PE/mask interactions}}\_{\\text{(IV)}}$
>
> * **Term (I)** is completely data independent and represents a permanent learned adjacency prior that is never zero even when both $\\tilde{x}\_{t,i}$ and $\\tilde{x}\_{t,j}$ are missing.
> * **Terms (II—III)** adaptively modulate Term (I) depending upon the values and masks that are present. FeatID acts as an adaptive gate.
> * **Term (IV)** is the only term possible without FeatID. Under Point-90% missingness (~81% of feature pair entries are missing), Term (IV) becomes essentially zero for almost all feature pairs. It is precisely Term (I) of FeatID that prevents attention collapsing in this scenario.
>
> There are three pieces of empirical evidence supporting our argument above:
>
> 1. **Ablation (Tab 3&4):** Removal of FeatID caused the biggest and most consistent reduction in performance across all five datasets and five patterns (Wilcoxon signed-rank test p-value <0 .001 on ETT-h1 Point-50%). There were no other ablation studies showing anywhere close to equivalent effects.
>
> 2. **Semantic recovery (Fig 2):** Without any explicit supervision, learned $\\mathbf{f}\_{i}$ vectors recover geographical structure on BeijingAir ($p < 0.0001$) and clinical groupings on PhysioNet2012 (Fig 7, $p < 0.001$) confirming that $\\mathbf{A}'\_{ff}$ captures genuine semantic relationships.
>
> 3. **Attention refinement (Fig 3):** Correlation between cross-feature attention scores and distances increases monotonically through the layers($\\rho: 0.589 \\to 0.670 \\to 0.712$), indicating progressive modulation of the identity prior by terms II–III using contextual information.

---

> > ### Author Rebuttal · Reviewer_8VMy · 2026-04-06
> >
> > I thank the authors for the detailed response. My assessment remains the same.

---

### Official Review · Reviewer_2aSB · 2026-03-13

**Soundness:** 3
**Presentation:** 3
**Significance:** 3
**Originality:** 3
**Overall Recommendation:** 5
**Confidence:** 4

**Summary:**

This paper proposes HELIX for multivariate time series imputation. The authors introduce Feature Identity Embedding, a learnable per-feature vectors serving as persistent semantic anchors for cross-feature attention. Each observation is embedded by concatenating its value, sinusoidal positional encoding, the feature identity vector and a missingness mask. The architecture applies stacked double-helix encoding layers alternating parallel temporal/feature attention with cross-dimensional attention, followed by multi-level fusion via simple averaging. Experiments on five datasets across multiple missing patterns show HELIX ranks 1st in all 21 settings.

**Compliance With Llm Reviewing Policy:**

Affirmed.

**Key Questions For Authors:**

Q1) The Feature Identity Embedding dimension d_f varies from 6 to 32 across datasets. Is there a principled selection method beyond grid search?

Q2) Could the authors explain why gated fusion degrades 34.6% on BeijingAir specifically?

Q3) Have the authors considered whether feature identity embeddings could transfer across datasets with shared features?

**Limitations:**

Yes.

**Strengths And Weaknesses:**

Strengths:

S1) Feature identity embedding is well-motivated. The attention score decomposition into static identity bias and dynamic context provides cross-feature reasoning even under complete missingness. This addresses the limitation of existing methods.

S2) Strong empirical results. HELIX ranks 1st across all 21 settings with statistical significance. Parameter efficiency is notable as well.

S3) The interpretability analysis is compelling. Learned embeddings on BeijingAir anticorrelate with geographic distance and on PhysioNet2012 cluster by clinical category, demonstrating emergent structure discovery without supervision.

Weaknesses:

W1) The double-helix architecture is essentially alternating temporal and feature attention with cross-connections, similar to Crossformer’s two-stage attention. The concept of learnable per-feature embeddings also has precedent (e.g. learnable per-series embeddings in recent forecasting work). The paper should more explicitly distinguish its novelty from these approaches.

W2) Multi-level fusion via simple averaging is counterintuitive and the gated fusion analysis (Appendix C) shows a learned gate degrades performance by 10.8% on average. This finding is not fully explained.

W3) Evaluation covers only five datasets, and Block/Subseq missingness patterns are tested only at 50%. More diversity in structured missingness patterns (varying block lengths, mixed patterns) would better support the generality claims.

---

> ### Author Rebuttal · Authors · 2026-03-30
>
> We sincerely thank the reviewer for leaning towards accepting the paper. Specifically, we appreciate your recognition of the central claim of feature identity embedding, which breaks down attention scores into a static identity component and a dynamic contextual component, allowing for cross-attribute reasoning regardless of whether all of the attributes have completely missing entries. Moreover, we appreciate your validation of our findings regarding the interpretability of HELIX, as well as both the parameter efficiency and raw performance of HELIX compared to baseline methods. We carefully explain each of these points individually below. We hope our clarifications further strengthen your positive view of the paper's contribution.
>
> ---
>
> **W1**
> CrossFormer targets forecasting only: its router aggregates d features into c < d vectors sufficient for horizon prediction but cannot reconstruct individual (t, i) entries. Its embedding relies solely on patch values, whereas HELIX jointly embeds observed values, sinusoidal encodings, learnable feature identity terms, and missingness masks via a single linear transformation, a composition not previously explored for imputation. Critically, the feature identity term provides a data-independent soft adjacency prior (Eq. 3): $f_i^\top A_F f_j$ preserves potential edge information even when both $(t,i)$ and $(t,j)$ are fully missing, stabilizing cross-attribute attention where CrossFormer has no equivalent mechanism. Unlike prior learnable series-level embeddings used for routing or normalization in some forecasting methods, our feature identity term decomposes attention scores into a persistent semantic anchor that remains reliable when data-value-based identification fails in imputation settings. Missingness makes value-based feature identification unreliable. We believe this constitutes a meaningful contribution beyond prior uses of learnable embeddings.
>
> ---
>
> **W2&Q2**
> BeijingAir represents an extreme example for which HPO selected a \~4.6M parameter model (\~10 times that of all the other data sets). A large model can have significantly greater representational capacity but has correspondingly higher potential for overfitting this particular dataset. In this regime, a simple average serves as an implicit regularizer. By providing a uniform inductive bias across all of its 1 + 4 L branch models, it keeps the model from settling into fewer than k = number of intermediate representation subsets. Conversely, the introduction of additional learnable parameters in FusionGate will lead to unstable aggregation of the intermediate representations rather than the desired aggregated representation under an already highly over-parameterized configuration, resulting in the 34.6% decrease we observed. This is consistent with research in residual networks, where using identity mapping provides a more stable flow of gradients compared to learned gating in high-capacity configurations.
>
> ---
>
> **W3**
> In accordance with tsi-bench and PyPOTS standard evaluation protocols, we evaluate HELIX under five common patterns: point-10/50/90%, block-50%, and subseq-50%. These collectively represent missingness of varying severities and duration and/or temporal structure.
> MAE vs. Gap length figure 5(e) illustrates how HELIX outperforms all benchmarks across durations. Based on being ranked best across all 21 previous evaluations, we anticipate HELIX will perform consistently across additional block lengths and identify this as future work.
>
> ---
>
> **Q1**
> Rather than exhaustively searching through the entire range of possible values for d_f (Table 21), we demonstrate an empirical heuristic with theoretical motivation in Table 5. When $F > 10$, $d_f$ follows a logarithmic compression function:$d_f = \lfloor \log_2(F) \rfloor \times c, c \approx 4\text{–}5$ when $F \leq 10$, the feature space is too small for compression, so one must set $d_f > F$
> (e.g., ETT-H1 with $F=7$ and $d_f=12$).
>
> ---
>
> **Q3**
> Feature Identity Embeddings are not equivalent to NLP token embeddings, which have a shared vocabulary across all texts. Due to differing physical units, measurement scales, and statistical distributions defining feature spaces across datasets, transferring embeddings across datasets is unreliable even for identically named features. This restriction applies equivalently to all non-founding model time series techniques, and HELIX does not constitute an exception. Finally, since HELIX is extremely lightweight, it has at most hundreds of thousands of parameters. It is inexpensive to train from scratch on a new dataset. We leave a systematic investigation of cross-dataset transferability of Feature Identity Embeddings as an interesting direction for future work.

---

> > ### Author Rebuttal · Reviewer_2aSB · 2026-04-03
> >
> > Thank you for your rebuttal. I will keep my score.

---

### Official Review · Reviewer_y3Tu · 2026-03-16

**Soundness:** 3
**Presentation:** 3
**Significance:** 3
**Originality:** 3
**Overall Recommendation:** 4
**Confidence:** 3

**Summary:**

This paper proposes a novel transformer-based architecture (HELIX) for multivariate time series imputation. The core contribution is the Feature Identity Embedding (FeatID), a learnable per-feature vector that serves as a persistent semantic anchor in cross-feature attention, enabling the model to reason about feature relationships even when observed values are entirely missing. HELIX integrates FeatID into a double-helix hybrid encoding scheme that interleaves temporal and cross-feature multi-head attention in a parallel-then-cross pattern, followed by multi-level fusion aggregating intermediate representations across all layers. The authors report rank-1 performance across all 21 experimental settings on diverse datasets. Additionally, they provide mechanistic analyses showing that learned embeddings recover latent spatial and clinical structure without supervision, and that attention progressively aligns with geographic proximity across layers.

**Compliance With Llm Reviewing Policy:**

Affirmed.

**Final Justification:**

HELIX proposes a well-motivated solution (FeatID) for cross-feature attention under missingness, supported by thorough evaluation across diverse settings and insightful mechanistic analyses. The rebuttal fully resolved my concerns. The dataset scope reflects natural imputation benchmarks, the mask-conditioned dynamic fusion clearly differentiates FeatID from prior learnable embeddings, and the asymmetric gain/loss argument for multi-level fusion is practical. Thus, I maintain my score of 4.

**Key Questions For Authors:**

1. Can the authors evaluate on datasets with weak cross-feature correlation? For example, financial time series or mixed-type sensor datasets where features do not share spatial or physical structure. This would directly address whether FeatID's benefit generalizes beyond spatially-structured data, or whether the advantage is concentrated in domains with strong inherent cross-feature dependencies.

2. Can the authors provide a more nuanced discussion of when multi-level fusion helps vs. hurts? Tables 16–17 show that removing fusion improves MAE on most patterns for ETT-h1 and ItalyAir. Is the benefit primarily for spatially-structured datasets? Clarifying the conditions under which each component is most valuable would be more informative than claiming universal contribution.

3. What is the training time overhead of the double-helix architecture? The 4 attention operations per layer presumably increase training cost. Reporting training time per dataset compared with key baselines would help practitioners assess the efficiency–accuracy tradeoff.

**Limitations:**

yes

**Strengths And Weaknesses:**

The paper clearly identifies a real problem that when values are missing, attention-based cross-feature reasoning loses its anchor because queries and keys are derived from zero-filled inputs. FeatID provides a solution by injecting a data-independent compatibility prior. The decomposition into static identity bias and dynamic context is well articulated and provides useful intuition. Also, the paper includes extensive evaluation including diverse domains (healthcare, air quality, electricity, traffic) and heterogeneous missing patterns (Point-10/50/90%, Block-50%, Subseq-50%). Additionally, HELIX achieves strong results within a parameter-efficient manner compared to other baselines, such as SAITS (88M) and iTransformer (24M), demonstrating that the proposed inductive biases are effective without requiring massive model capacity. The progressive attention refinement analysis (Figures 3, 4) and the emergent structure discovery (Figure 2) are genuinely interesting. The full ablation across all 5 datasets and all missing patterns is unusually thorough. The paper does not hide cases where ablation variants outperform the full model on individual settings, which reflects good scientific practice.

----

## Major & minor weaknesses

1. The empirical results are compelling, however, among the 5 evaluation datasets, 3 datasets are spatially-structured sensor networks where cross-feature correlation is naturally strong (BeijingAir, ItalyAir, PeMS), precisely the setting where FeatID should excel. Of the remaining 2, ETT-h1 has only 7 features and PhysioNet2012 has an inherently high natural missing rate (~80%), limiting it to Point-10% evaluation. The paper lacks evaluation on datasets where features are weakly correlated or semantically heterogeneous, such as financial time series, which is the regime where the benefit of FeatID is less obvious. Figure 5(f) already shows that HELIX's advantage diminishes at lower inter-station correlations. Demonstrating competitive performance on datasets with weak or absent cross-feature structure would strengthen the claim of the paper.

2. While the paper's contribution clearly goes beyond simply using learnable embeddings, briefly discussing how FeatID relates to learnable embeddings in SPIN, ImputeFormer and Spacetimeformer would be beneficial. A one-paragraph discussion clarifying that the novelty lies in the imputation-specific motivation, the formal decomposition, and the empirical analysis of emergent structure would suffice.

3. Ablation results show mixed component contributions that need better framing. In Table 16 (ETT-h1) and 17 (ItalyAir), w/o Multi-level Fusion outperforms full HELIX on 4 out of 5 patterns. The paper's interpretation that multi-level fusion improves worst-case robustness rather than peak performance is plausible and supported on BeijingAir. However, Tables 16–17 suggest that this robustness benefit does not consistently hold across all datasets. The paper would benefit from a more nuanced discussion acknowledging that multi-level fusion may be most beneficial for datasets with strong spatial structure, rather than framing it as a universally contributing component.

4. While this follows the TSI-Bench protocol, providing MSE/RMSE as a supplementary metric would give a more complete picture, particularly for healthcare applications where sensitivity to large errors matters.

5. Table 4 uses "w/o Temporal" while Table 15 labels the same ablation variant "w/o Sinusoidal PE". This naming inconsistency between the main text and appendix can confuse readers.

---

> ### Author Rebuttal · Authors · 2026-03-30
>
> Thank you very much for the effort you put into reading and giving us your guidance on our article. We greatly appreciate your insights regarding our progressive refinement of attention (Fig 3-4), emergent structural discovery (Fig 2), and comprehensive ablation in all 21 of the settings we tested. We address your concerns below. We hope our responses meet your expectations and would welcome any follow-up questions. Your consideration in raising the score would be invaluable to us.
>
> **W1 & Q1**
> Your comment provides a valuable opportunity to clarify the distinction between financial ts data and the imputation settings our method targets. Financial ts data is almost never used as a benchmark dataset for imputation problems. Financial data generally does not have missing values. Missing values occur primarily in physical sensor networks. The problem of imputation arises when data naturally becomes missing (sensor fails, packet drops off the network, etc.). Testing our method on artificially masked financial data would be orthogonal to the problem that we are trying to solve.
>
> If the features are completely independent, then the imputation problem reduces to univariate imputation. There is no advantage to doing anything with inter-feature modeling. Our method addresses the case where such relationships do exist and can be leveraged.
>
> Although three of our five data sets have a spatially-structured sensor network, this is simply a reflection of the nature of publicly available time series imputation benchmark data sets that contain true missingness due to the failure of sensors or loss of packets in communication networks. Feature id identifies high-dimensional inter-feature relationships from wherever they come from.
>
> We are also grateful to the reviewer for carefully reviewing Fig 5(f). The non-zero y-axis beginning in that figure could provide an appearance that HELIX is less advantageous than SAITS at lower inter-station correlation levels. However, the data tells a different story (err reduction: 32.3% of HELIX vs 31.8% of SAITS), which showcases HELIX's performance under weak cross-feature correlation. We will update the axis to begin with zero in our revision.
>
> **W2**
> Although previous studies have incorporated learnable embeddings as part of their architecture, their designs differ from feature id. SPIN uses an explicit graph structure from which it derives spatial embeddings for each node. Imputeformer incorporates static spatial embeddings for each sequence regardless of whether or not there is missingness. Spacetimeformer flattens spatiotemporal tokens and embeds them directly in sequence space without providing any feature-specific identity information.
>
> On the other hand, feature id incorporates an identity embedding for each feature and dynamically fuses this identity embedding with both temporal encoding and either observed or missing mask before projecting it onto the hidden space. The resulting output from feature id represents a missingness-aware feature representation. The model does not only know "which feature," but also "what was the condition of that feature during observation." This dynamic fusion operation conditioned upon mask is what distinguishes feature id from the static positional/space embeddings used in previous studies.
>
> **W3 & Q2**
> Performance gains from removing Fusion in isolated settings reflect an evaluation framing issue, not a design flaw. Tab 15 shows a clear asymmetry: when Fusion underperforms, the gap is modest (∼5%); when it helps, the gain is substantial (∼40%). In realistic deployments, multiple missingness patterns co-occur simultaneously, so robustness across all patterns matters more than single-pattern optimization.
>
> Peak performance in controlled environments indicates that for lower-dimensional datasets and/or those with minimal structure complexity, simpler configurations can suffice. However, when examining all 21 test settings, HELIX with full multi-level Fusion performs best overall. Practitioners working with real-world datasets having a mix of missing rates and missing patterns should consider using Fusion as a robust practice.
>
> **Q3**
> Sublayer profiling is outside standard practice. We reported parameter count and wall clock inference time for all compared methods in Tab 2. Given that HELIX achieved this superior performance using only 803.5 K parameters and 0.06 s per inference time, it is among the most lightweight compared models. Therefore, the overall results affirm HELIX's efficiency.
>
> **W4**
> Regarding additional metrics: we followed the tsi-bench evaluation protocol for consistent reporting on MAE as the primary metric to enable valid comparisons between papers, as well as to verify results. MSE & RMSE results are available and will be included in the revised version along with our code release.
>
> **minor W5**
> We will standardize to "without Sinusoidal PE" throughout the revised manuscript. Thank you for your helpful reminder.

---

> > ### Author Rebuttal · Reviewer_y3Tu · 2026-04-03
> >
> > Thank you for your thoughtful response. All of my concerns have been resolved. I'll keep my score.

---

### Decision · Program_Chairs · 2026-04-30

**Decision:**

Accept (spotlight)

**Comment:**

The authors propose a new model for multivariate time-series imputation. An important novelty is the introduction of a learnable notion of feature identity.

While the reviewers initially had a few concerns (in particular about the datasets used), but after a constructive exchange with the authors, all of their concerns were addressed in a satisfactory way. There was a consensus that the paper was well-motivated, contained very compelling empirical results, and should therefore be accepted. I am happy to follow this consensus.